# Differential translation of mRNA isoforms underlies oncogenic activation of cell cycle kinase Aurora A

Roberta Cacioppo[1]*, Hesna Begum Akman[1,2], Taner Tuncer[3], Ayse Elif Erson-Bensan[2], Catherine Lindon[1]*

[1]Department of Pharmacology, University of Cambridge, Cambridge, United Kingdom; [2]Department of Biological Sciences, Orta Dogu Teknik Universitesi, Ankara, Turkey; [3]Department of Biology, Ondokuz Mayis Universitesi, Samsun, Turkey

**Abstract** Aurora Kinase A (AURKA) is an oncogenic kinase with major roles in mitosis, but also exerts cell cycle- and kinase-independent functions linked to cancer. Therefore, control of its expression, as well as its activity, is crucial. A short and a long 3'UTR isoform exist for AURKA mRNA, resulting from alternative polyadenylation (APA). We initially observed that in triple-negative breast cancer, where AURKA is typically overexpressed, the short isoform is predominant and this correlates with faster relapse times of patients. The short isoform is characterized by higher translational efficiency since translation and decay rate of the long isoform are targeted by *hsa-let-7a* tumor-suppressor miRNA. Additionally, *hsa-let-7a* regulates the cell cycle periodicity of translation of the long isoform, whereas the short isoform is translated highly and constantly throughout interphase. Finally, disrupted production of the long isoform led to an increase in proliferation and migration rates of cells. In summary, we uncovered a new mechanism dependent on the cooperation between APA and miRNA targeting likely to be a route of oncogenic activation of human AURKA.

## eLife assessment

In this **important** study, the authors provide **compelling** evidence that the interplay between alternative polyadenylation (APA) of mRNA encoding Aurora Kinase A (AURKA) and *hsa-let-7a* miRNA governs AURKA protein levels. The authors show that short 3'UTR isoform of mRNA encoding AURKA is efficiently translated throughout the cell cycle, while the long 3'UTR isoform is suppressed by *hsa-let-7a* miRNA in a cell cycle-dependent manner. These findings delineate post-transcriptional mechanisms regulating AURKA expression that may be implicated in increase in AURKA protein that is frequently observed across a variety of cancers.

## Introduction

Aurora Kinase A (AURKA) is a critical positive regulator of the mitotic phase of the cell cycle (***Willems et al., 2018***). AURKA also plays additional cancer-promoting roles in cell proliferation, survival, migration, and cancer stem cell phenotypes, some of which in interphase and in a kinase-independent manner (***Naso et al., 2021***). AURKA expression follows a strict cell cycle-dependent pattern, with both protein and mRNA levels extremely low in $G_1$ phase, increasing in S phase, and peaking at $G_2$ phase until mitosis (***Cacioppo and Lindon, 2022***). High expression of AURKA is strongly associated with cancer progression, drug resistance, and poor prognosis, justifying why oncogenic AURKA represents a renowned target of anticancer drugs (***Du et al., 2021***), and making evident that oncogenic roles of AURKA are prompted by its highly sustained levels of expression.

**\*For correspondence:**
rc781@cam.ac.uk (RC);
acl34@cam.ac.uk (CL)

**Competing interest:** The authors declare that no competing interests exist.

AURKA overexpression in human cancers is known to be caused by elevated gene copy number, enhanced transcription, or increased protein stability. Dysregulation of translation is also linked to disease and contributions of dysregulated translation to cancer phenotypes are increasingly reported (*Kovalski et al., 2022*; *Modelska et al., 2015*). Despite some modest evidence suggesting that modulation of AURKA translation is relevant in disease (*Dobson et al., 2013*; *Lai et al., 2017*), control of AURKA expression at the level of translation is widely understudied compared with control of its transcription and mRNA processing (*Cacioppo and Lindon, 2022*). For example, it is not clear whether AURKA mRNA undergoes translational activation and/or inhibition during the cell cycle, and the precise timing, extent, or regulators of these processes remain unexplored.

The process of cleavage of the 3'end of precursor mRNAs (pre-mRNAs) and concomitant addition of a poly(A) tail represents one key event aiding the maturation of mRNAs, termed cleavage, and polyadenylation (C/P) (*Gruber and Zavolan, 2019*). The cleavage site is typically preceded by a polya-denylation signal (PAS), located 10–30 nucleotides (nt) upstream, and by UGUA and U-rich motifs, whereas it is typically followed by U- and GU-rich motifs. Altogether, these elements constitute the C/P site (*Gruber et al., 2016*). Most human pre-mRNAs contain multiple C/P sites (*Derti et al., 2012*), enabling alternative cleavage and polyadenylation (APA) and, thus, distinct expression of transcript isoforms for the same gene. A search using PolyA_DB (*Wang et al., 2018*) indicates the presence of two C/P sites with canonical PASs (AATAAA) on AURKA 3' untranslated region (3'UTR) (*Figure 1A*). This fostered our hypothesis that AURKA mRNA could be subjected to tandem 3'UTR APA, resulting in two 3'UTR isoforms that differ in length. It is currently unknown which AURKA PAS is preferentially used in which cellular context or whether a 3'UTR isoform switch can be modulatable.

APA is involved in most cellular processes and is often altered in cancer (*Gruber and Zavolan, 2019*). Clinically, human cancers are characterized by unique profiles of alternative 3'UTRs that can be exploited for classification of distinct cancer subtypes (*Singh et al., 2009*; *Wang et al., 2020*), and associations between 3'UTR shortening and poor prognosis (*Lembo et al., 2012*) or drug sensitivity *Xiang et al., 2018* have been detected. At the molecular level, a strong positive association between expression of short 3'UTRs, increased protein levels, and proliferative states has been frequently reported (*Sandberg et al., 2008*; *Mayr and Bartel, 2009*; *Masamha et al., 2014*; *Xia et al., 2014*; *Pieraccioli et al., 2022*). Such genome-wide 3'UTR shortening sustains cancer cell behavior by removing repressor sequence elements from the 3'UTR of oncogenic mRNAs, for example, microRNA (miRNA) binding sites (*Sandberg et al., 2008*; *Mayr and Bartel, 2009*; *Masamha et al., 2014*), or alternatively by inactivating tumor suppressors through suppression of their expression (*Lee et al., 2018*; *Park et al., 2018*).

The role of miRNAs in regulating cell cycle genes and the relevance of this regulation in cancer are well understood (*Bueno and Malumbres, 2011*; *Ghafouri-Fard et al., 2020*). Few miRNAs have been pointed to as regulators of AURKA mRNA but, importantly, reported cases of miRNA targeting of AURKA occur in those cancers where AURKA overexpression is a promoting factor or a marker of poor prognosis (*Fadaka et al., 2020*; *Zhang et al., 2020*; *Yuan et al., 2019*; *Ma et al., 2015*). Regardless, none of these studies consider the existence of distinct AURKA 3'UTR isoforms in their experimental design of targeting assessment. The *hsa-let-7* miRNA family comprises 11 closely related genes that map in chromosomal regions that are typically deleted in human tumors and, given their pathogenic downregulation in cancer, they are classified as tumor suppressors (*Bueno and Malumbres, 2011*; *Johnson et al., 2007*). Roles for *hsa-let-7a* in breast tumor growth and metastasis have been proposed (*Thammaiah and Jayaram, 2016*; *Shi et al., 2020*) and a correlation between *hsa-let-7a* expression and clinical variables has been detected in triple-negative breast cancer (TNBC) (*Avery-Kiejda et al., 2014*; *Turashvili et al., 2018*).

AURKA was classified within the TNBC subtype with the highest median index of 3'UTR shortening events (*Wang et al., 2020*; *Akman et al., 2015*), and also undergoes 3'UTR shortening in poor-prognosis patients of breast and lung cancer (*Lembo et al., 2012*). Importantly, AURKA overexpres-sion in TNBC represents a marker of early recurrence, poor prognosis, and shorter overall survival (*Xu et al., 2013*; *Jalalirad et al., 2021*). However, the correlation between AURKA PAS usage, protein expression, and pathological cell behavior has not been explored for this or other biological contexts, nor at the molecular level. In this study, we uncover a molecular mechanism leveraging the cellular ratio of APA isoforms and their different translational program during the cell cycle to control acquisi-tion of AURKA oncogenic potential.

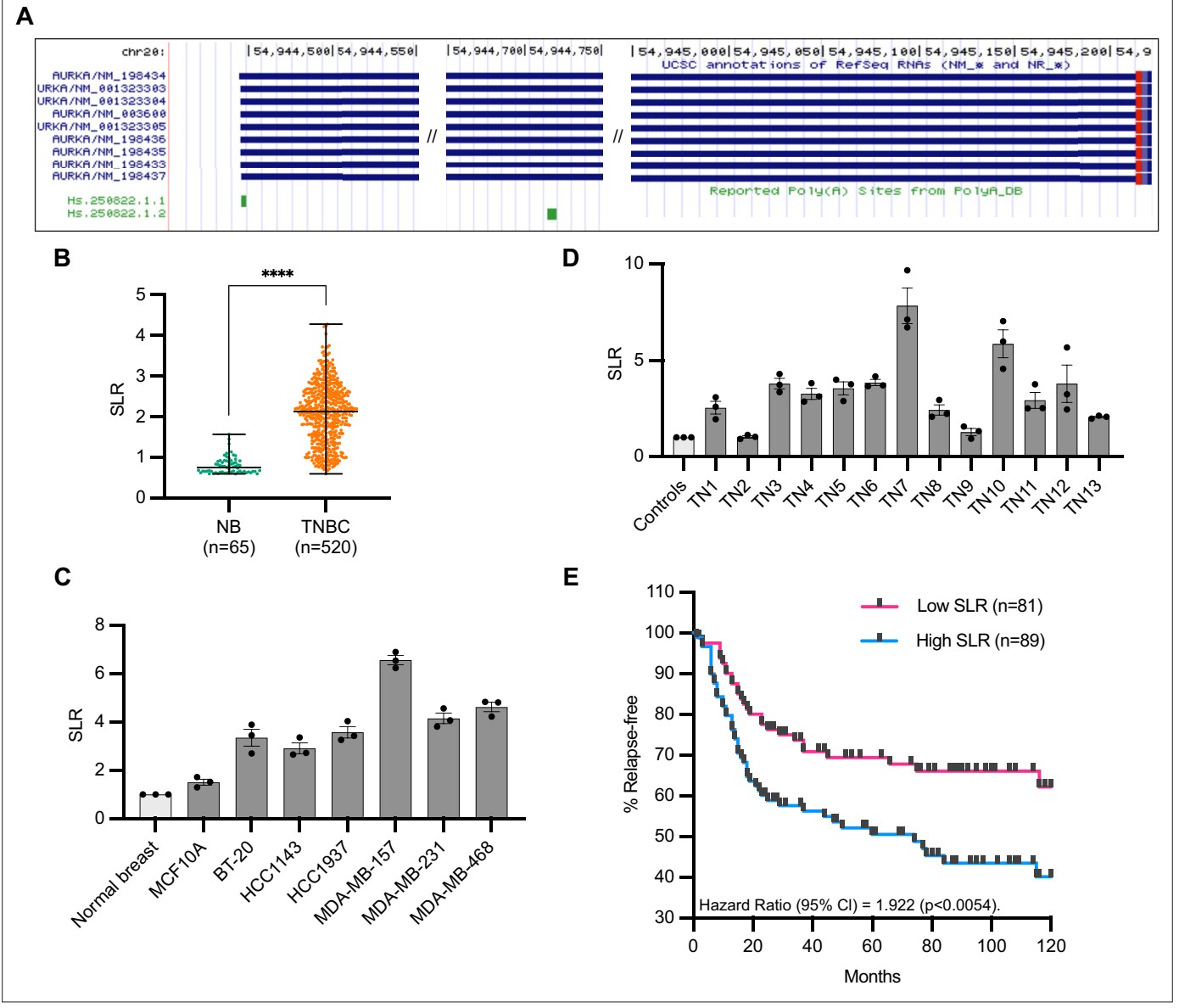

**Figure 1.** Increased short/long ratio (SLR) of Aurora Kinase A (AURKA) alternative polyadenylation (APA) isoforms in triple-negative breast cancer (TNBC). (**A**) AURKA transcript isoforms (USCS Genome Browser). AURKA gene is located on (-) strand. (**B**) Median and range of SLR values for AURKA 3'UTR obtained using APADetect. Mann–Whitney test; ****p<0.0001. (**C**), (**D**) RT-qPCR analysis of SLR of AURKA 3'UTR in TNBC cell lines (**C**) and patient samples (**D**). SDHA used as reference gene. TN, tissue number. (**E**) Relapse-free survival rates of TNBC patients with high (highest 25%) or low (lowest 25%) AURKA SLRs. p-value determined by log-rank test.

The online version of this article includes the following source data for figure 1:

**Source data 1.** Numerical data for graphs.

## Results

### Increased short/long ratio of AURKA APA isoforms in TNBC

In a preliminary study using the APADetect in silico tool, we analyzed publicly available microarray data to identify changes in AURKA 3'UTR isoform abundance in tissues (*Akman et al., 2015*). 520 comparable datasets for TNBC samples came from GSE31519 (*Rody et al., 2011*) and 65 histologically normal epithelium and cancer-free prophylactic mastectomy patients were used: 32 from GSE20437 (*Graham et al., 2010*), 12 from GSE9574 (*Tripathi et al., 2008*), 7 from GSE3744 (*Richardson et al., 2006*), 6 from GSE6883 (*Liu et al., 2007*), 5 from GSE26910 (*Planche et al., 2011*),

and 3 from GSE21422 (*Kretschmer et al., 2011*). The analysis revealed increased short/long ratio (SLR) of AURKA 3'UTR isoforms in TNBC compared to normal breast tissues (*Figure 1B*). Higher SLR was confirmed by RT-qPCR in multiple TNBC cell lines (*Figure 1C*). Furthermore, RT-qPCR analysis of normal and TNBC patient cDNAs from Origene Breast Cancer cDNA array IV (BCRT504) also showed higher AURKA SLR in TNBC samples compared to normal (*Figure 1D*). In addition, the shortening of AURKA 3'UTR correlated with faster relapse times in TNBC patients (clinical data from *Rody et al., 2011*; *Figure 1E*). These results therefore suggest a potential oncogenic role of AURKA APA in breast cancer worth further investigations.

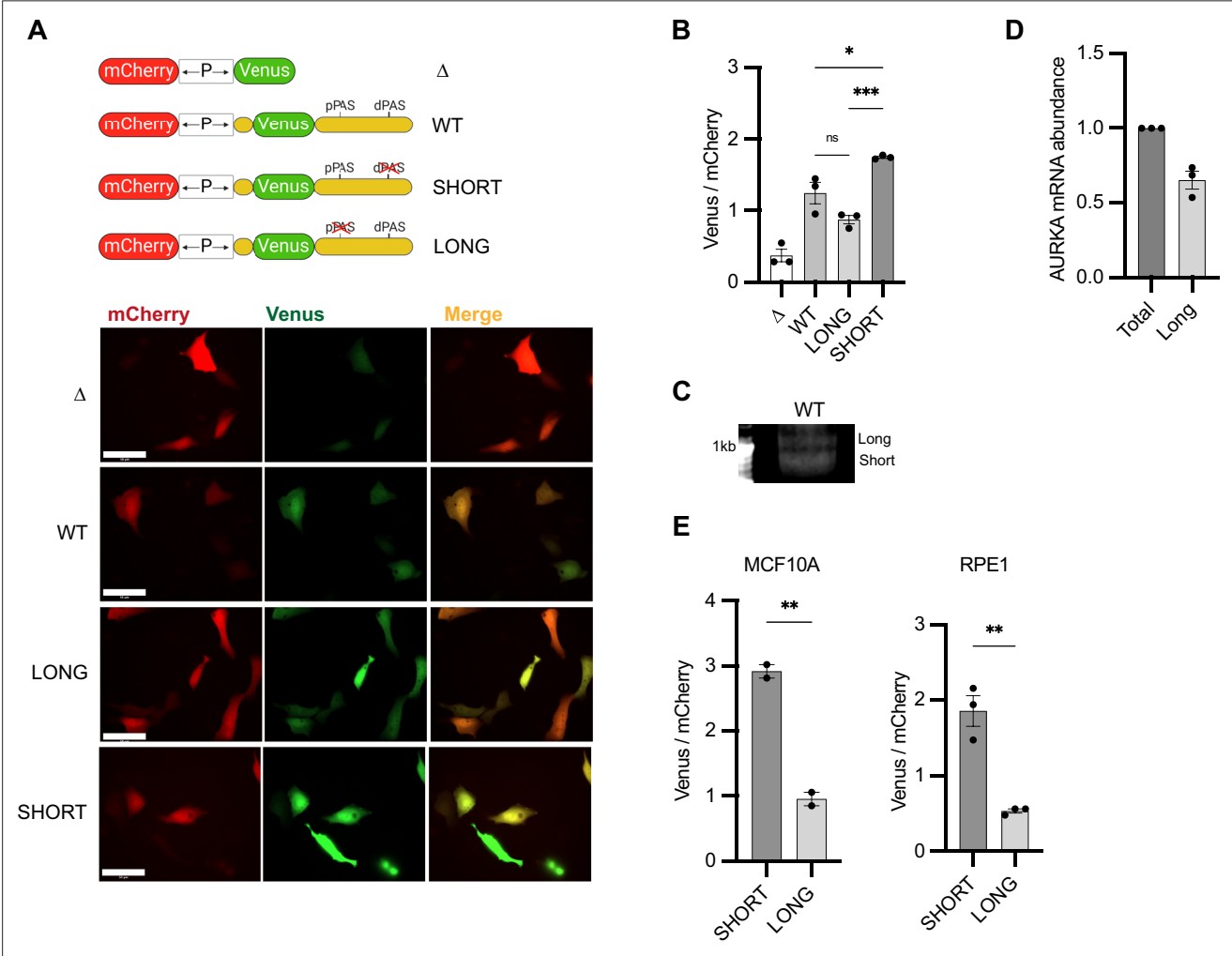

**Figure 2.** Aurora Kinase A (AURKA) shows 3'UTR isoform-dependent protein expression. (**A**) Top: UTR-dependent protein expression reporters. Venus coding sequence (CDS) is flanked by AURKA 5'UTR and 3'UTR, WT or polyadenylation signal (PAS)-mutated. Bottom: representative snapshots of transfected U2OS cells. Scale bar 50 µm. (**B**) Mean and SEM of median Venus/mCherry mean fluorescence intensity (MFI) ratios from transfected U2OS cells from three biological replicates. n ≥ 129 cells per condition. Ordinary one-way ANOVA with Tukeys multiple-comparisons test. (**C**) 3'RACE of endogenous AURKA APA isoforms. (**D**) RT-qPCR of endogenous AURKA short/long ratio (SLR) in U2OS cells. Long isoform abundance plotted as fold change over total AURKA mRNA. 18S rRNA used as reference target. (**E**) Same as (**B**) but in MCF10A (left) and RPE1 (right) cells. n ≥ 55 cells per condition. Unpaired *t*-test. ns, not significant; *p<0.05; **p<0.01; ***p<0.001.

The online version of this article includes the following source data and figure supplement(s) for figure 2:

**Source data 1.** Raw gel/blot images.

**Source data 2.** Numerical data for graphs.

**Figure supplement 1.** Validation of bidirectional reporter.

**Figure supplement 1—source data 1.** Numerical data for graphs.

## AURKA shows 3′UTR isoform-dependent protein expression

To probe AURKA APA isoform-dependent protein expression, we developed a single-cell expression sensor suitable for experiments in live cell. The construct independently expresses Venus and mCherry fluorescent proteins via a constitutive bi-directional promoter (*Figure 2A*). The coding sequence (CDS) of Venus is flanked by AURKA UTRs, whereas that of mCherry lacks regulatory regions and is therefore used to normalize for transfection efficiency. To test for APA-sensitive expression, we alternatively mutated the distal (d) or proximal (p) PAS on the reporter 3′UTR, to generate different 3′UTR isoforms (SHORT and LONG, respectively). Constructs lacking AURKA UTRs (Δ) and expressing AURKA wild-type UTRs (WT) were used as controls.

We initially assessed the efficiency of the promoter bidirectionality. Correlation between Venus and mCherry expression was strongly maintained at the level of both fluorescence intensity (*Figure 2— figure supplement 1A*) and mRNA abundance (*Figure 2—figure supplement 1B*). Promoter strength was, however, not equal in both directions since fewer copies of mCherry mRNA were transcribed compared to Venus mRNA (*Figure 2—figure supplement 1B*), despite mCherry fluorescence intensity being generally higher than that of Venus (*Figure 2A and B*, see Δ). We further assessed that mRNA of both mCherry and Venus was stable over time (*Figure 3B*, see Δ), and fluorescence of both proteins was stable over time and over different cell cycle stages (*Figure 2—figure supplement 1C and D*). Considering the short maturation time and long half-life of both Venus and mCherry proteins (*Shaner et al., 2004*; *Nagai et al., 2002*), the assay allows to reliably measure effects of UTRs on reporter protein levels at any given time and regardless of cell cycle phases. As positive control of our assay, we recapitulated the higher protein expression from the short 3′UTR APA isoform of CDC6 mRNA, which has previously been observed (*Akman et al., 2012*; *Figure 2—figure supplement 1E*).

Addition of AURKA UTRs to Venus CDS significantly increased protein expression (*Figure 2A and B*), likely due to the role of 5′UTR in facilitating translation (*Hinnebusch et al., 2016*). We found that the SHORT reporter generates significantly more protein compared to the LONG (*Figure 2A and B*). Moreover, similar protein expression levels from the WT and LONG reporters suggest that AURKA WT 3′UTR is processed with a preference for dPAS in U2OS cells. Accordingly, we could detect both endogenous AURKA APA isoforms in U2OS cells by 3′RACE (*Figure 2C*) and confirmed by RT-qPCR that AURKA long isoform is prevalent (~60% of total AURKA mRNA) (*Figure 2D*). The different SLR observed between U2OS cells, normal breast tissues, and TNBC cell lines and tissues (*Figure 1B–D*) indicates that AURKA 3′UTR isoform prevalence is dependent on cell type. In addition, the quantitative difference in reporter protein expression by the isoforms also varied among cell types, suggesting cell-specific regulation (*Figure 2B and E*, SHORT vs. LONG). In sum, these results provide evidence for the first time of a role for APA in controlling AURKA protein expression.

## AURKA APA isoforms are translated with different efficiency

We next investigated the basis of the different protein expression between AURKA APA isoforms. Following transfection of U2OS cells with the constructs in *Figure 2A*, we first quantified the abundance of reporter mRNA isoforms (*Figure 3A*). We then assessed the isoforms decay rate by quantifying reporter mRNAs at multiple time points following arrest of transcription by actinomycin D (ActD) (*Figure 3B*). We observed that while Venus mRNA lacking UTRs was highly stable, reporter mRNA levels decreased at faster rate when carrying UTRs (*Figure 3B*), indicating that the assay reports on UTR-dependent effects on mRNA stability. Both the abundance and stability of the SHORT and LONG reporter isoforms were similar (*Figure 3A and B*). We additionally found that the two endogenous AURKA 3′UTR isoforms also have similar decay rates, albeit decaying at a higher rate compared to the reporter mRNAs (*Figure 3C*), suggesting that features present in AURKA CDS might influence mRNA stability (*Narula et al., 2019*).

Because the reporter APA isoforms share similar abundance and stability, we wondered whether they undergo different translational regulation instead. To this aim, we adapted a biochemical translation efficiency (TE) assay from *Williams et al., 2022*, which required addition of a 3XFlag tag to the N-terminus of Venus (FlagVenus) in our reporter constructs, and called this nascent chain immunoprecipitation (NC IP) assay (*Figure 3D*). We first assessed that addition of the 3XFlag tag did not alter Venus expression (*Figure 3E*). In our NC IP assay, anti-Flag beads were used to immunoprecipitate nascent FlagVenus chains from ribosomes stalled by treatment with cycloheximide (CHX). Ribosome-mRNA complexes were eluted from the IP-immobilized nascent chains using puromycin,

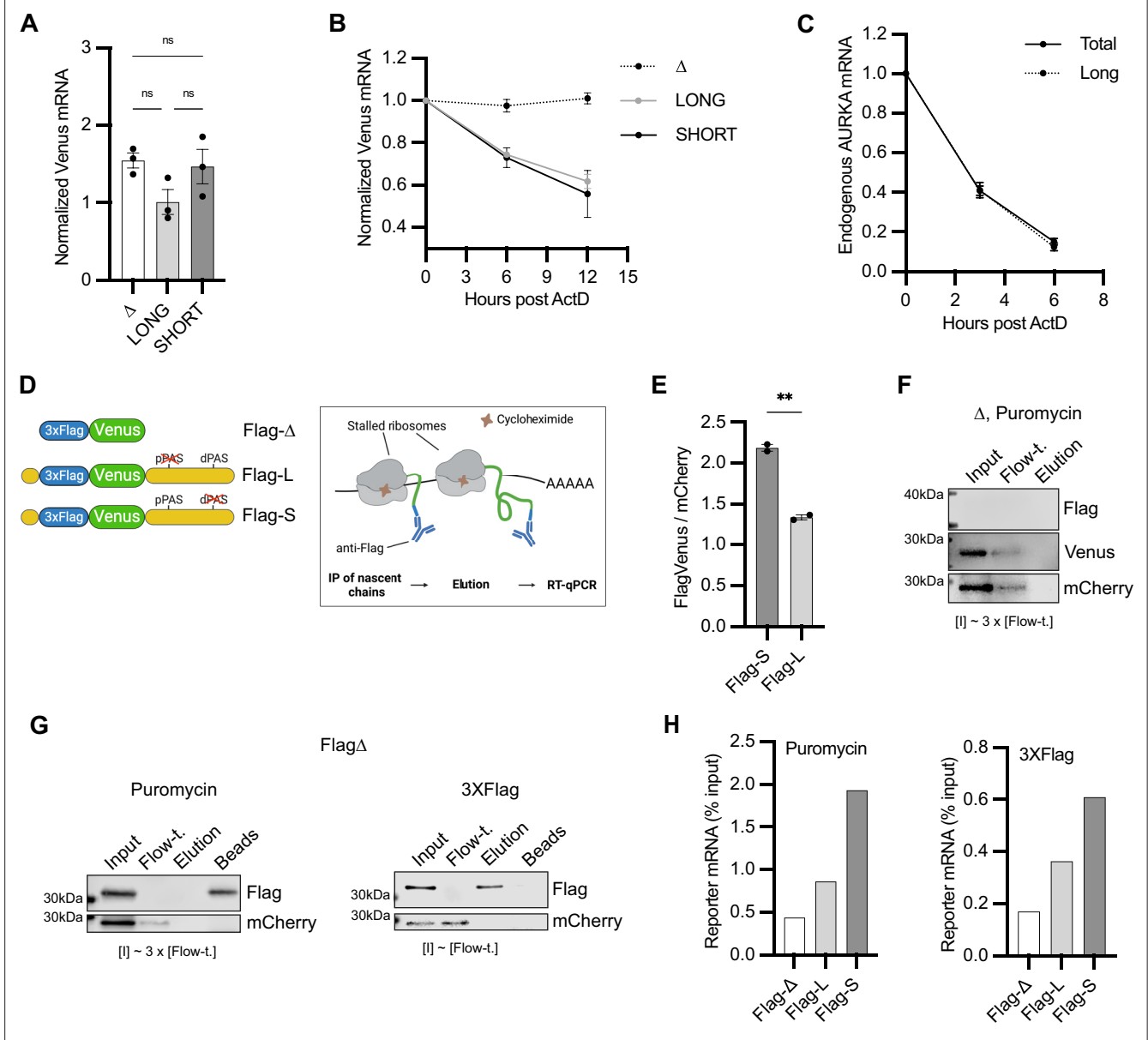

**Figure 3.** Aurora Kinase A (AURKA) alternative polyadenylation (APA) isoforms are translated with different efficiency. (**A**, **B**) RT-qPCR of reporter mRNAs abundance (**A**) and decay rate (**B**) from transfected U2OS cells. mCherry mRNA used as reference target. Ordinary one-way ANOVA with Tukey's multiple-comparisons test; ns, not significant. (**C**) Decay rate of endogenous AURKA mRNA as in (**B**). 18S rRNA used as reference target. Abundance of long isoform plotted as fold change over total AURKA mRNA. (**D**) Design of the nascent chain immunoprecipitation (NC IP) reporters and assay. (**E**) Mean and SEM of median FlagVenus/mCherry mean fluorescence intensity (MFI) ratios from transfected U2OS cells from two biological replicates. n ≥ 160 cells per condition. Unpaired $t$-test; **$p < 0.005$. (**F**), (**G**) Immunoblots of NC IP fractions using Δ (**F**) or Flag-Δ (**G**) reporter. mCherry used as negative control. (**H**) RT-qPCR of eluted reporter mRNAs. Results representative of three biological replicates.

The online version of this article includes the following source data for figure 3:

**Source data 1.** Numerical data for graphs.

**Source data 2.** Raw gel/blot images.

which causes release of nascent chains from ribosomes (*Aviner, 2020*); RNA was then purified from the elution fraction and quantified by RT-qPCR to provide a measure of the amount of reporter mRNA undergoing translation. All fractions were then blotted for FlagVenus and mCherry (negative control) proteins to monitor their presence at different steps of the experiment. As expected, elution with puromycin retained FlagVenus on the beads, whereas mCherry, as well as untagged Venus, are lost in

the flow-through (*Figure 3G, left, and F*). No reporter mRNA could be detected in the elution fraction when untagged Venus was used (data not shown). Alternatively, purified 3XFlag peptide was used to elute the nascent chain-ribosome-mRNA complexes from the beads following IP (*Figure 3G*, right). RT-qPCR quantification of the FlagVenus nascent chain-cognate reporter mRNAs revealed about twice more copies of Flag-S mRNA compared to Flag-L mRNA, regardless of the elution method (*Figure 3H*). This indicates that Flag-S mRNA is translated with higher efficiency than Flag-L mRNA. These results show that APA controls AURKA protein expression mainly via differential translational regulation of the 3′UTR isoforms.

## Translation rate of AURKA APA isoforms follows different cell cycle periodicity

Given the known cell cycle-dependent expression of AURKA (*Cacioppo and Lindon, 2022*), we tested whether differential translational efficiency of AURKA mRNA isoforms might contribute to this regulation. To avoid perturbation of translation provoked by classical cell cycle synchronization methods (*Anda and Grallert, 2019*), we used a live-cell fluorescence-based translation rate measurement assay in conjunction with a CDK2 activity sensor (*Spencer et al., 2013*) for in silico cell cycle synchronization. We developed our assay of 'translation rate imaging by rate of protein stabilization' (TRIPS) based on a previously introduced reporter system (*Han et al., 2014*; *Tanenbaum et al., 2015*). Our bidirectional promoter construct was modified to express superfolder GFP (sfGFP) fused to a mutated *Echerichia coli* dihydrofolate reductase (DHFR-Y100I) destabilizer domain (DHFR-sfGFP), which is continuously degraded unless the stabilizer molecule trimethoprim (TMP) is added. Addition of TMP leads to an increase of sfGFP signal over time and, given the sfGFP short maturation time (*Pédelacq et al., 2006*), the accumulation rate of sfGFP reflects DHFR-sfGFP protein synthesis rate (*Figure 4A*, left, *Figure 4—figure supplement 1A*). The ratio of the median of single-cell mCherry-normalized sfGFP signals at 2 hr to that at 0 hr of TMP treatment was therefore used as read-out for bulk translation rate. In accordance with our assay being designed to measure translation rate, sfGFP signal could not increase under TMP treatment in the presence of translation inhibitor CHX (*Figure 4B*). We also ensured that the increase in sfGFP signal was TMP-dependent (*Figure 4—figure supplement 1B*) and that TMP treatment affected neither mCherry expression (*Figure 4—figure supplement 1C*) nor DHFR-sfGFP mRNA abundance (*Figure 4C*). To probe translation rate at different cell cycle phases, we used the CDK2 activity sensor (*Spencer et al., 2013*) stably expressed in our U2OS cell line (U2OS$^{CDK2}$) (*Figure 4—figure supplement 1A*) and called this assay 'cell cycle-dependent TRIPS' (C-TRIPS) (*Figure 4A*, right).

To test the translation rate of the individual AURKA APA isoforms, we flanked DHFR-sfGFP CDS with AURKA PAS-mutated UTRs (TRIPS-L, TRIPS-S) (*Figure 4A*, left). We found that our TRIPS assay could recapitulate the difference in translation efficiency of the isoforms previously observed (*Figures 3H and 4D*, left). Importantly, expression of the CDK2 activity sensor did not affect cellular translation as the different translation rate of AURKA 3′UTR isoforms could be reproduced in U2OS cells lacking the sensor (*Figure 4—figure supplement 1D*). Following measurements of bulk translation rates (*Figure 4D*, left), we then binned single-cell translation rate values into three intervals of CDK2 activity (*Figure 4D*, right). Results of our C-TRIPS assay revealed that, while TRIPS-Δ is translated constantly during the cell cycle, translation rate of TRIPS-L is regulated in the cell cycle. This isoform showed lower translation rate in G$_1$ and S and an enhanced rate at G$_2$, consistent with the increase in both AURKA mRNA and protein levels that occurs in preparation for mitosis (*Cacioppo and Lindon, 2022*). By contrast, TRIPS-S was translated constantly through the cell cycle and at a maximal rate already in G$_1$ (*Figure 4D*, right), indicating that this isoform is insensitive to cell cycle regulation of AURKA translation rate.

Furthermore, we quantified abundance of endogenous AURKA APA isoforms at different stages of the cell cycle by performing RT-qPCR following synchronization in G$_1$/S, G$_2$, or M phases (*Figure 4E*). The expected changes in AURKA mRNA abundance following each treatment represent a positive control for the synchronization. However, abundance of the long isoform changed quite concomitantly with changes in total AURKA mRNA levels (*Figure 4F*). This suggests that the same ratio of 3′UTR isoforms is rather maintained throughout the cell cycle and that AURKA APA is not cell cycle regulated.

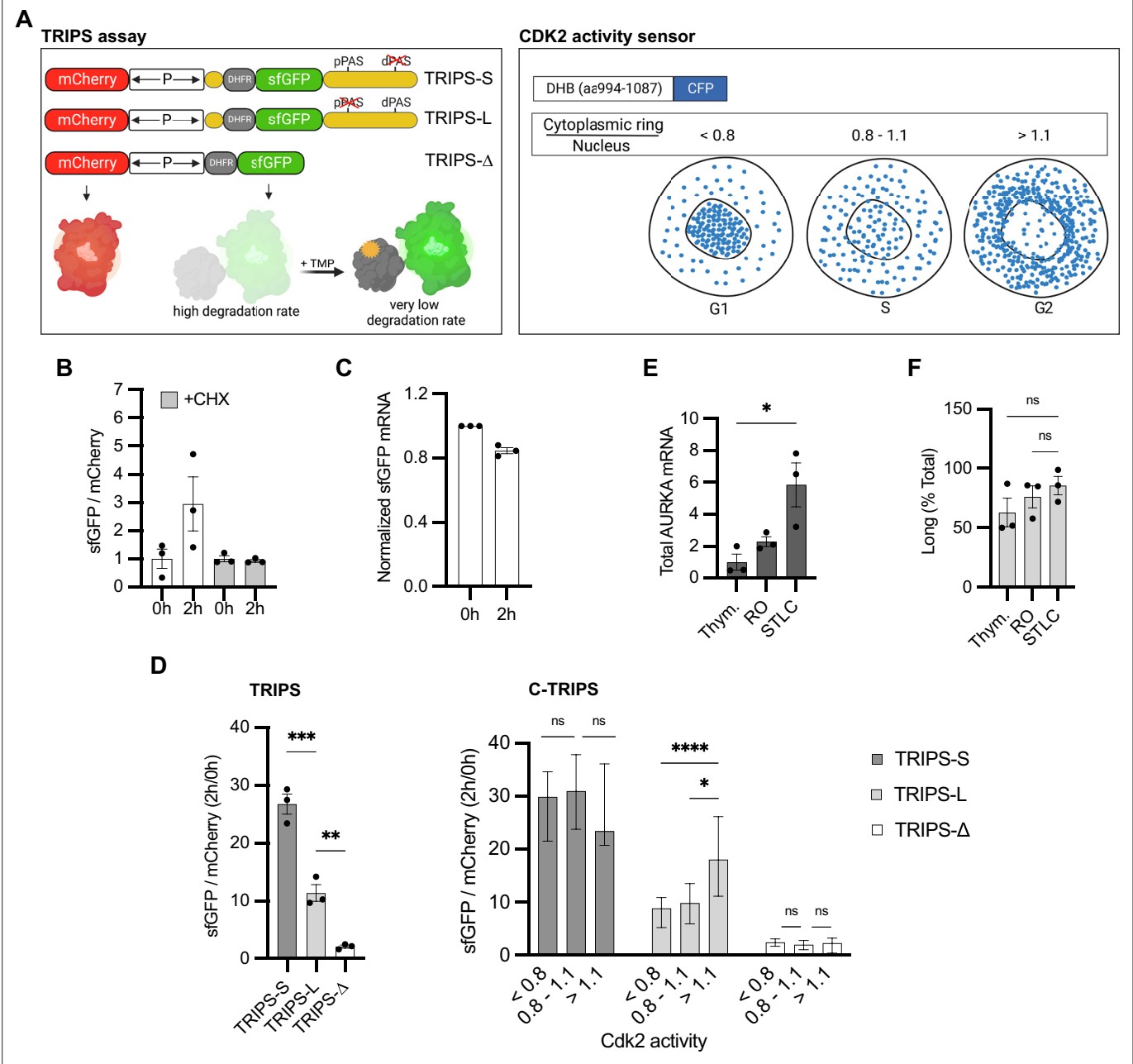

**Figure 4.** Translation rate of Aurora Kinase A (AURKA) alternative polyadenylation (APA) isoforms follows different cell cycle periodicity. (**A**) Design of the translation rate imaging by rate of protein stabilization (TRIPS) reporters and assay (left) and CDK2 activity sensor (right). (**B**) Mean and SEM of median sfGFP/mCherry mean fluorescence intensity (MFI) ratios from U2OS cells transfected with TRIPS-Δ and imaged at 0 hr and 2 hr of 50 µM trimethoprim (TMP) treatment, with or without 0.1 mg/ml cycloheximide (CHX), from three biological replicates. Baseline at 0 hr. n ≥ 55 cells per condition. (**C**) RT-qPCR of sfGFP mRNA from U2OS cells transfected with TRIPS-Δ, at 0 hr and 2 hr of 50 µM TMP treatment. mCherry mRNA used as reference target. RNA extracts at 0 hr of treatment used as reference sample. (**D**) TRIPS (left) and C-TRIPS (right) assays in transfected U2OS$^{CDK2}$ cells. n ≥ 200 cells per condition. Left: mean and SEM from three biological replicates. Right: median and 95% CI of pooled data from left. Kruskal–Wallis with Dunnett's multiple-comparisons test. (**E**) RT-qPCR of endogenous AURKA mRNA in U2OS cells. 18S rRNA used as reference target. Baseline at G$_1$/S. (**F**) Endogenous AURKA long isoform as in (**E**) plotted as percentage of total AURKA mRNA. (**D**) Left, (**E**, **F**) Ordinary one-way ANOVA with Tukey's multiple-comparisons test. ns, not significant; *p<0.05; **p<0.01; ***p<0.0005; ****p<0.0001.

The online version of this article includes the following source data and figure supplement(s) for figure 4:

**Source data 1.** Numerical data for graphs.

**Figure supplement 1.** Validation of TRIPS assay.

**Figure supplement 1—source data 1.** Numerical data for graphs.

These results not only provide strong, independent validation of our finding that elements present in 5′ and 3′ UTR of AURKA enable translational activation but additionally indicate that elements present on the long 3′UTR might account for its different pattern of translation during interphase as lack of these on the short 3′UTR allow escape from cell cycle phase-dependent translation.

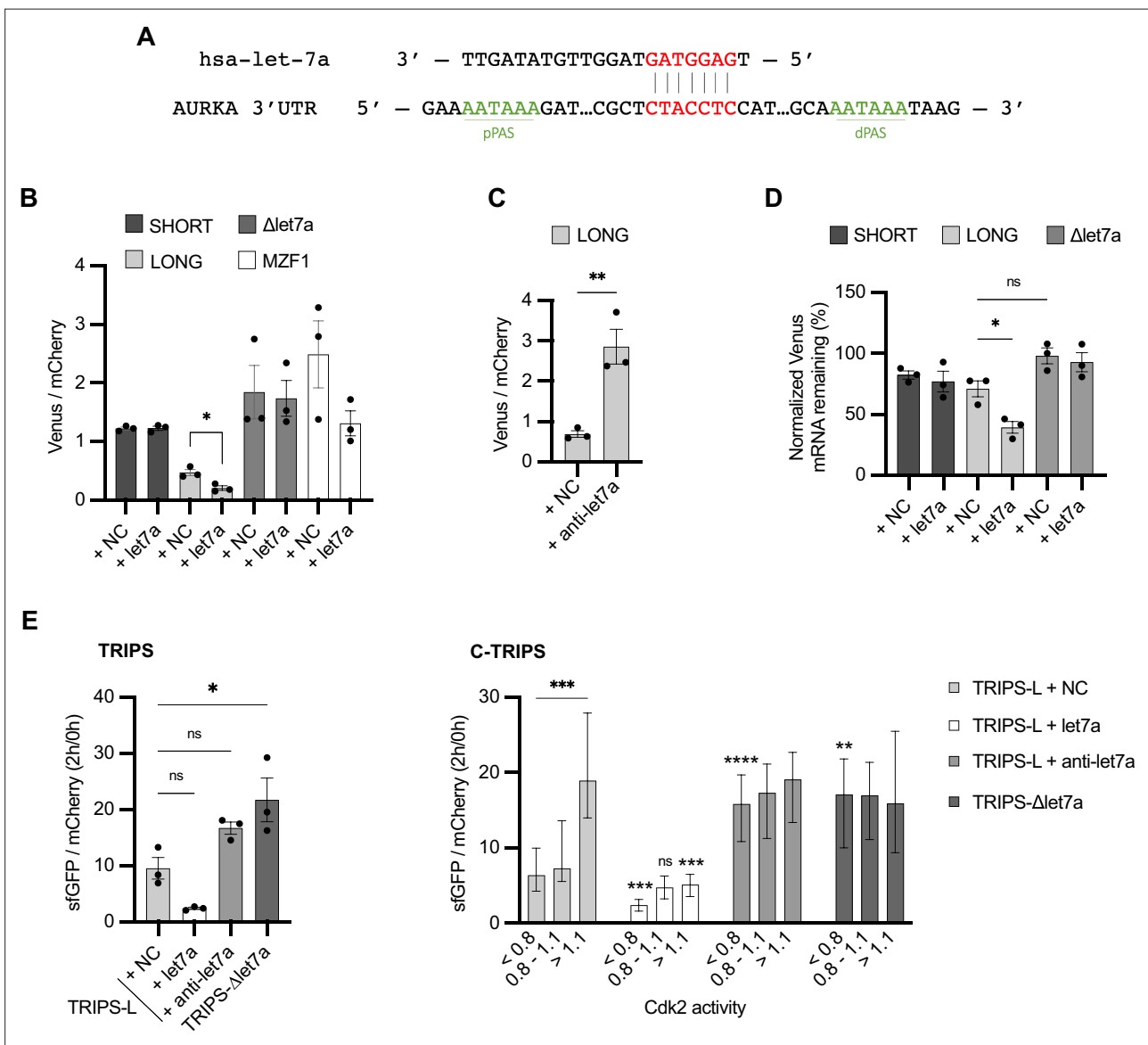

**Figure 5.** Translational periodicity of long 3′UTR isoform is regulated by *hsa-let-7a* miRNA. (**A**) Complementarity of *hsa-let-7a* binding to Aurora Kinase A (AURKA) 3′UTR. (**B**) Mean and SEM of median Venus/mCherry mean fluorescence intensity (MFI) ratios from U2OS cells co-transfected with 250 nM *hsa-let-7a* or a negative control (NC) miRNA from three biological replicates. n ≥ 182 cells per condition. Unpaired *t*-test. (**C**) Same as (**B**) but co-transfecting 300 nM *anti-let-7a* or NC. n ≥ 94 cells per condition. Unpaired *t*-test. (**D**) RT-qPCR of reporter mRNAs abundance from U2OS cells transfected as (**B**), at 8 hr of 10 µg/ml ActD. mCherry mRNA used as reference target. Ordinary one-way ANOVA with Tukey's multiple-comparisons test. (**E**) Translation rate imaging by rate of protein stabilization (TRIPS) (left) and C-TRIPS (right) assays in transfected U2OS^CDK2 cells. n ≥ 162 cells per condition Left: mean and SEM from three biological replicates. Ordinary one-way ANOVA with Dunnett's multiple-comparisons test vs. NC. Right: median and 95% CI of pooled data from left. Kruskal–Wallis with Dunnett's multiple-comparisons test vs. NC of the respective phase. ns, not significant; *p<0.05; **p<0.01; ***p<0.001; ****p<0.0001.

The online version of this article includes the following source data for figure 5:

**Source data 1.** Numerical data for graphs.

## Translational periodicity of long 3'UTR isoform is regulated by *hsa-let-7a* miRNA

Among known post-transcriptional regulators, miRNAs are widely recognized as molecular regulators of both mRNA stability and translation (*Jonas and Izaurralde, 2015*). We interrogated miRDB (https://mirdb.org/) (*Chen and Wang, 2020*) to search for miRNAs that could be involved in the differential regulation of the two AURKA mRNA isoforms and selected *hsa-let-7a* miRNA (*Figure 5A*) given its widely established tumor-suppressor role of in TNBC. We assessed the *hsa-let-7a* targeting of AURKA 3'UTR by co-transfecting our AURKA UTR-dependent protein expression reporters (*Figure 2A*) with *hsa-let-7a* or a negative control miRNA that does not have any target in the human genome. As positive control of the assay, we cloned Myeloid Zinc Finger1 (MZF1) 3'UTR downstream Venus CDS in our Δ reporter and could reproduce the previously reported targeting of MZF1 3'UTR by *hsa-let-7a* (*Tvingsholm et al., 2018*; *Figure 5B*). Protein expression from the LONG reporter mRNA was reduced by *hsa-let-7a*, whereas that from the SHORT mRNA and from a LONG mRNA that lacks the *hsa-let-7a* binding site (Δlet7a) was not (*Figure 5B*). Also, the loss of *hsa-let-7a* targeting was sufficient to increase protein expression from the LONG reporter mRNA (compare LONG + NC vs. Δlet7a + NC). To confirm that altered expression was due to the lack of *hsa-let-7a* targeting and not an effect of the mutation itself, we also observed an increase in protein expression when we co-transfected our LONG reporter and an inhibitor of *hsa-let-7a* (*anti-let7a*) (*Figure 5C*).

In order to assess the role of *hsa-let-7a* in controlling decay rate of the target mRNA, we next co-transfected our Venus reporters and either *hsa-let-7a* or negative control miRNA and quantified reporter mRNA abundance after 8 hr of ActD treatment. We found that stability of the LONG reporter mRNA was significantly reduced by *hsa-let-7a*, whereas that of the SHORT reporter mRNA was unaltered. Additionally, mutation of the *hsa-let-7a* binding site slightly increased reporter mRNA stability (compare LONG + NC vs. Δlet7a + NC) (*Figure 5D*).

We then performed our C-TRIPS assay co-transfecting the TRIPS-L reporter and *hsa-let-7a* or control miRNA and found that *hsa-let-7a* reduced both bulk translation rate (*Figure 5E*, left) and translation rate at all interphase stages (*Figure 5E*, right) of the long 3'UTR. Furthermore, we asked whether loss of *hsa-let-7a* targeting is sufficient to cause loss of translational regulation of the long isoform during the cell cycle. For this, we performed the C-TRIPS assay using a TRIPS-L reporter with mutations in the *hsa-let-7a* binding site (TRIPS-Δlet7a) or, alternatively, co-transfecting the TRIPS-L reporter and the *hsa-let-7a* inhibitor *anti-let7a*. Interestingly, in both cases, loss of *hsa-let-7a* targeting only increased translation rate in $G_1$ and S, but not $G_2$ (*Figure 5E*, right), suggesting that the targeting in $G_2$ is not likely to occur unless in conditions of excess *hsa-let-7a*.

In conclusion, our results show that *hsa-let-7a* only silences AURKA long 3'UTR isoform by both promoting mRNA degradation and reducing translation rate, and that *hsa-let-7a* targeting is responsible for the cell cycle-dependent translational regulation of AURKA long 3'UTR isoform.

## Increased AURKA short/long ratio is sufficient to disrupt cell behavior

Having established that APA plays a role in regulating AURKA expression, we tested the idea that AURKA APA directly contributes to cancer cell behavior by performing genome editing to alter AURKA APA in wild-type U2OS cells. We used Cas9$^{D10A}$-mediated double-nicking strategy and mutated the endogenous dPAS on *AURKA* 3'UTR with the aim of silencing expression of AURKA long 3'UTR isoform (*Figure 6A*). Two mutant clones with disrupted dPAS site were obtained (ΔdPAS#1, ΔdPAS#2) (*Figure 6—figure supplement 1A*) and were used for subsequent functional analyses. Qualitative assessment of AURKA 3'UTR isoforms ratio in these clones by 3'RACE found the long 3'UTR isoform to be undetectable in the mutated cell lines (*Figure 6B*), indicating that the genetic editing successfully prevents usage of the dPAS site for cleavage and polyadenylation.

We then examined expression of AURKA in the mutated cell lines by immunoblot of extracts from cell populations enriched for the $G_1$/S phase of the cell cycle, where AURKA expression is the lowest in unmodified cells. We observed AURKA was expressed at higher levels in ΔdPAS#1 and ΔdPAS#2 cells compared to WT cells (*Figure 6C*, *Figure 6—figure supplement 1B*). AURKA expression in $G_1$/S was reduced in the mutated cell lines when treated with CHX, indicating that translation of the short isoform is active in this phase (*Figure 6D*). Because AURKA overexpression is a common feature of cancer, we interrogated the mutated cell lines for changes in cancer-relevant behavior. Consistent with a role of AURKA overexpression in accelerating the cell cycle and favoring cell proliferation, we found

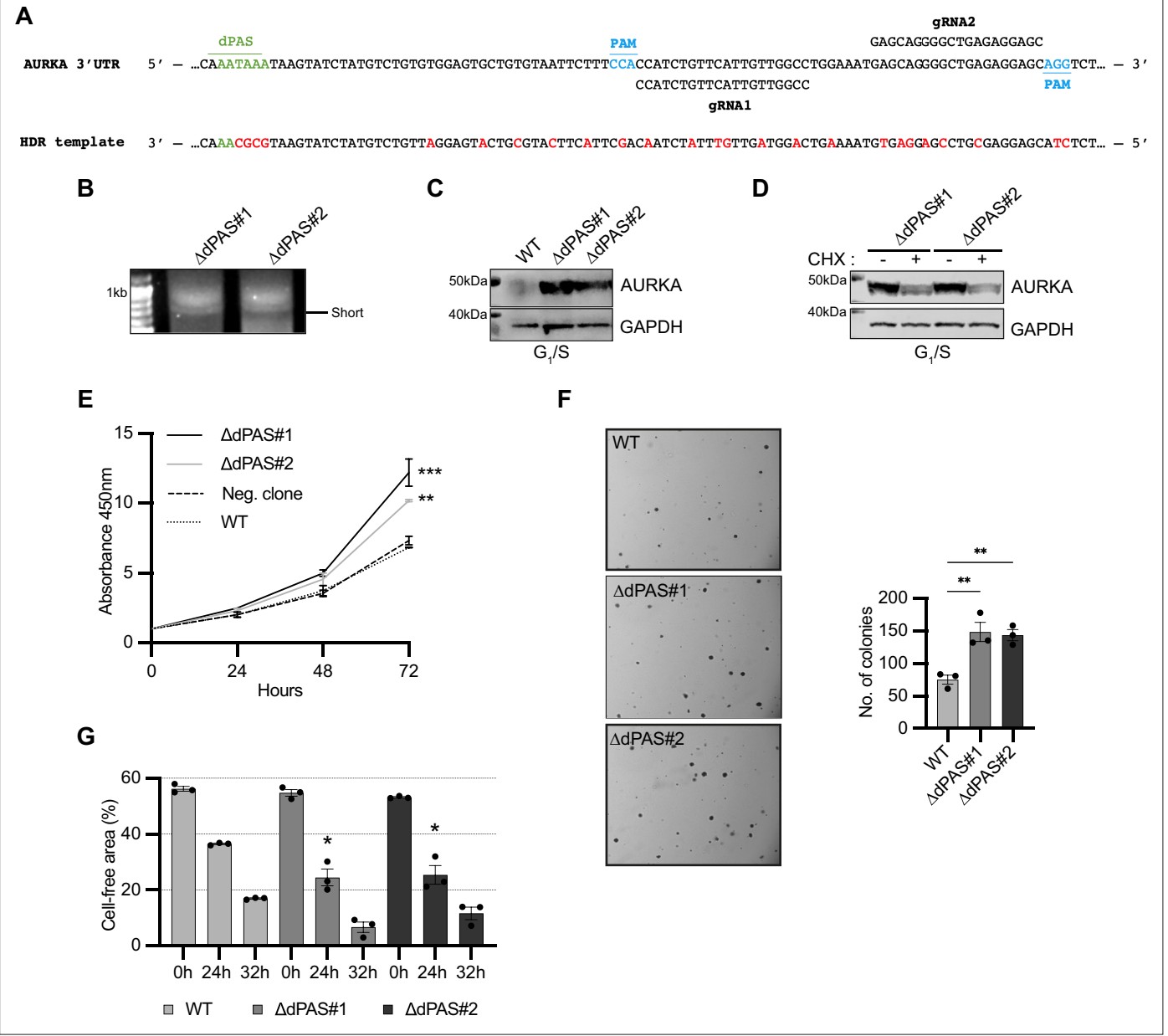

**Figure 6.** Increased Aurora Kinase A (AURKA) short/long ratio is sufficient to disrupt cell behavior. (**A**) Design of CRISPR editing. Nucleotide substitutions in red. (**B**) 3'RACE of endogenous AURKA alternative polyadenylation (APA) isoforms. (**C**), (**D**) Western blot after $G_1$/S enrichment (**C**) and with or without 6 hr treatment with 0.1 mg/ml cycloheximide (CHX) (**D**). Blots representative of three biological replicates. (**E**) CCK8 assay. A nonparental WT U2OS cell line used as negative control. Ordinary one-way ANOVA with Dunnett's multiple-comparisons test vs. WT; **$p<0.005$; ***$p<0.001$. (**F**) Left: representative images of cells grown in soft agar. Right: mean number of clones and SEM of three biological replicates. (**G**) Measurement of migration rate. (**F**, **G**) Ordinary one-way ANOVA with Dunnett's multiple-comparisons test vs. WT; *$p<0.05$; **$p<0.01$.

The online version of this article includes the following source data and figure supplement(s) for figure 6:

**Source data 1.** Raw gel/blot images.

**Source data 2.** Numerical data for graphs.

**Figure supplement 1.** Validation of mutated cell lines.

**Figure supplement 1—source data 1.** Raw gel/blot images.

a higher rate of proliferation of ΔdPAS#1 and ΔdPAS#2 cells compared to WT cells using the CCK8 assay to measure metabolic activity (*Figure 6E*). Additionally, we assessed the ability of anchorage-independent growth, which closely correlates with tumorigenicity in animal models by growing cells in soft agar. ΔdPAS#1 and ΔdPAS#2 cells resulted more capable to survive and grow in the absence of anchorage to their neighboring cells compared to WT cells (*Figure 6F*). AURKA also regulates organization of microtubules required for cellular migration and also enhances migration of tumor cells through several pathways. For example, AURKA activates the Cofilin-F-Actin pathway leading to breast cancer metastases (*Willems et al., 2018*). We were therefore prompted to assess the motility of ΔdPAS#1 and ΔdPAS#2 cells in a 2D cell migration assay. Our result shows a higher rate of migration of ΔdPAS#1 and ΔdPAS#2 cells compared to WT cells (*Figure 6G*).

In summary, our results show that AURKA overexpression caused by a disruption in the SLR of APA isoforms in favor of the short isoform contributes to cancer-like cell behavior.

## Discussion

In this study, we describe for the first time a molecular mechanism involving the dysregulation of APA and the differential targeting of AURKA APA isoforms by *hsa-let-7a*, a tumor suppressor miRNA, that is sufficient for the oncogenic activation of AURKA at the post-transcriptional level. We also shed light on the cell cycle-dependent regulation of AURKA translation and introduce novel and improved methods for measurements of post-transcriptional gene expression of individual mRNAs of interest.

As a consequence of tandem 3′UTR APA, a short and a long 3′UTR isoform are generated for AURKA mRNA. The SLR of these isoforms is cell-type defined and is not dependent on the cell cycle, indicating that cell-type-specific/cell cycle-independent factors are involved in establishing the SLR. This is unexpected given that periodic regulation of APA may be a characteristic of many cell cycle genes (*Dominguez et al., 2016*). Because protein expression differs from the two isoforms, AURKA SLR is a crucial element defining AURKA expression levels. Ours and other studies detected increased SLR of AURKA APA isoforms in TNBC and found this correlates with worse disease-free survival (*Wang et al., 2020*; *Akman et al., 2015*). Here, we reveal insights into the molecular basis for the correlation by showing that the increased and cell cycle-independent translation rate of the short isoform can lead to marked overexpression of AURKA in interphase. Our results support the hypothesis that deregulation of expression by disruption of APA is sufficient to drive AURKA oncogenic properties, such as promoting increased proliferation and migration rate. Whether this is also sufficient to drive cancer-cell transformation remains to be explored. It has been known for many years that overexpression of AURKA induces mitotic defects, aneuploidy, as well as acceleration of the cell cycle, epithelial-to-mesenchymal transition and migration (*Willems et al., 2018*; *Zhou et al., 1998*), whilst the cellular background for AURKA's transforming potential is also of importance (*Asteriti et al., 2010*). Nonetheless, the significance of AURKA overexpression specifically in $G_1$ for the exertion of potential oncogenic functions in this phase represents an emerging field of research (*Naso et al., 2021*; *Abdelbaki et al., 2020*; *Bertolin and Tramier, 2020*). It is possible that normal AURKA functions are exerted at low levels of expression in $G_1$ become oncogenic at high levels of expression. This would not be surprising given how the roles played by AURKA in $G_1$ revolve around regulation of transcription, mitochondria fitness, and cellular metabolism.

Our work does not research the cause of the disrupted APA in TNBC. This could be concomitant to a global 3′UTR shortening, for example, due to altered expression in cancer of C/P factors (*Gruber and Zavolan, 2019*; *Wang et al., 2020*), or of their regulators (*Pieraccioli et al., 2022*), or even altered RNA Pol II elongation dynamics (*Mitschka and Mayr, 2022*). However, these phenomena have not been extensively explored in TNBC (*Miles et al., 2016*). Alternatively, disrupted APA could represent an AURKA gene-specific phenomenon due to the presence of single-nucleotide polymorphisms (SNPs) on the 3′UTR either on the C/P site or in proximity that could affect PAS choice by the C/P machinery. Finally, an mRNA-dependent mechanism whereby the short 3′UTR isoform itself regulates AURKA protein function or localization throughout or at specific phases of the cell cycle, licensing oncogenic advantage, should not be excluded (*Mitschka and Mayr, 2022*).

The periodicity of AURKA expression is an important requirement for correct progression through the cell cycle. For example, ubiquitin-mediated proteolysis is a critical pathway for irreversible AURKA inactivation, which must occur for ordered transition to interphase following mitosis (*Abdelbaki et al., 2020*; *Lindon et al., 2015*). One important but unanswered question is whether and to what

extent translation regulation accounts for the increase and decrease of AURKA expression during the cell cycle. Implementation of our TRIPS assay, which measures protein synthesis rate independent of changes in mRNA abundance, allowed detection of active regulatory mechanisms of translation occurring at different cell cycle phases. This new evidence integrates well with the notion that transcriptional, post-transcriptional, and post-translational mechanisms all combine to provide *AURKA* gene with a characteristic pattern of expression (*Cacioppo and Lindon, 2022*). Our work shows that translation of AURKA is regulated by *hsa-let-7a* miRNA. Because we could still detect active protein synthesis from the long isoform under *hsa-let-7a* overexpression using our TRIPS assay, and we could also immunoprecipitate tagged nascent chains from the long isoform in U2OS cells, where *hsa-let-7a* is expressed, it is unlikely that *hsa-let-7a* blocks translation at the level of initiation, but it rather slows down the rate of translation elongation. This is in accordance with a study showing that *hsa-let-7a* co-sediments with actively translating polyribosomes (*Nottrott et al., 2006*), a generally proved mechanism of miRNA action (*Tat et al., 2016*). In addition, because we observed that *hsa-let-7a* can also control the decay rate of the long isoform, it is possible that a reduction in translation elongation rate may be required to mediate degradation of the mRNA (*Biasini et al., 2021*). We also show that the differential *hsa-let-7a* targeting through the cell cycle is a mechanism responsible for AURKA periodic translational control. Based on evidence from our C-TRIPS assays to assess the temporal *hsa-let-7a* targeting of the long isoform at different phases of the cell cycle, and on the published evidence that *hsa-let-7a* levels are constant during the cell cycle of human cancer cells as well as untransformed fibroblasts (*Grolmusz et al., 2016*), we propose that (i) *hsa-let-7a* targeting is productive in $G_1$ and S, and is therefore responsible for the low AURKA translation rate in these phases; and (ii) the targeting is not occurring in $G_2$ phase, except in excess of *hsa-let-7a*, possibly because *hsa-let-7a* overexpression saturates sequestering factors that prevent its binding to the long isoform. Further investigations will be required to understand this mechanism in more detail.

The characterization of gene-specific post-transcriptional dynamics is desirable for a complete understanding of gene expression regulation. Here, we have developed transient single-cell and biochemical assays to rapidly study mRNA-specific gene expression in a way that measures post-transcriptional events exclusively. Importantly, the assays can be used to test the effect of regulators such as therapeutic miRNAs or drugs on protein expression, mRNA processing, and translation of selected genes.

In conclusion, our study reveals a strong cooperation between APA and miRNA targeting in controlling gene expression dynamics of AURKA and its oncogenic potential. It also provides a workflow to assess the role of mRNA-specific post-transcriptional processing and regulators. Our work additionally highlights a molecular mechanism that could represent an actionable target of RNA-based therapeutics.

## Materials and methods

### In silico analysis of APA in TNBC

Publicly available CEL files and associated metadata of microarray results were downloaded from NCBI Gene Expression Omnibus (GEO) repository. APADetect tool (*Akman et al., 2015*) was used to detect and quantify APA events in TNBC patients and normal breast tissue. CEL files of Human Genome U133A (HGU133A, GPL96) and U133 Plus 2.0 Arrays (HGU133Plus2, GPL570) were analyzed to identify intensities of probes that were grouped based on poly(A) site locations extracted from PolyA_DB (*Zhang et al., 2005*). Mean signal intensities of proximal and distal probe sets for AURKA were calculated and used as indicators of 'short' and 'long' AURKA 3′UTR isoforms' abundance. The ratio of proximal probe set mean (S) to the distal probe set (L) is defined as short to long ratio (SLR). SLR values were subjected to significance analysis of microarrays (SAM), as implemented by the TM4 Multiple Array Viewer tool (*Saeed et al., 2003*), for statistical significance after log normalization.

### Molecular cloning

The following UTR sequences were obtained by gene synthesis (Genewiz from Azenta Life Sciences, European Genomics Headquarters, Germany): AURKA WT 5′UTR and 3′UTR (769 bp) (NM_003600.4), MZF1 3′UTR (NM_003422.3), AURKA individual PAS-mutated 3′UTRs (AATAAA>AATCCC). CDC6 3′UTRs were from *Akman et al., 2012*. For Δ reporter, mCherry and Venus ORFs were inserted into

the MCSs of *pBI-CMV1* (631630, Clontech, TakaraBio). *WT*, *SHORT* and *LONG* reporters were generated by assembly (NEBuilder HiFi DNA Assembly Cloning Kit, E5520S, NEB) of AURKA 5′UTR, Venus ORF and AURKA wt, short (dPAS-mutated) or long (pPAS-mutated) 3′UTR, and insertion into *pBI-CMV1-mCherry*. *CDC6_L*, *CDC6_S*, and *MZF1* reporters were generated by insertion of CDC6 long or short 3′UTR, and MZF1 3′UTR downstream Venus CDS in Δ reporter. *LONG-Δlet7a* was generated by site-directed mutagenesis of *LONG* reporter with the following forward and reverse primers: 5′-CACGCACCATTTAGGGATTTGCTTG-3′ and 5′AGCACGTGTTCCTATTTTTCACACTC-3′. *Flag-Δ* was generated by insertion of 3XFlag-Venus ORF into *pBI-CMV1-mCherry*. *Flag-S* and *Flag-L* reporters were generated by assembly of AURKA 5′UTR, 3XFlag-Venus ORF and AURKA short (dPAS-mutated) or long (pPAS-mutated) 3′UTR, and insertion into *pBI-CMV1-mCherry*. For *TRIPS* reporters, the DHFR-sfGFP ORF was PCR amplified from *pHR-DHFRY100I-sfGFP-NLS-P2A-NLS-mCherry-P2A_Emi1 5' and 3'UTR* plasmid, a gift from Ron Vale (Addgene plasmid #67930), and inserted into *pBI-CMV1-mCherry* (*TRIPS-Δ*) or assembled with AURKA 5′UTR and AURKA wt 3′UTR (*TRIPS-WT*), short 3′UTR (*TRIPS-S*), or long 3′UTR (*TRIPS-L*) before insertion. *TRIPS-Δlet7a* was generated by site-directed mutagenesis of *TRIPS-L* reporter with the primers above. NEB 5-alpha Competent *E. coli* (High Efficiency) (C2987I, NEB) was used.

## Cell lines and drug treatments

Human U2OS, U2OS$^{CDK2}$, BT20, HCC1143, HCC1937, MDA-MB-157, MDA-MB-231, and MDA-MB-468 cell lines were cultured in DMEM (41966029, Thermo Fisher) supplemented with 10% FBS (F9665, Sigma), 200 µM GlutaMAX-1 (35050061, Thermo Fisher), 100 U/ml penicillin (15140122, Thermo Fisher), 100 µg/ml streptomycin (15140122, Thermo Fisher), and 250 ng/ml fungizone (15290026, Thermo Fisher) at 37°C with 5% $CO_2$. U2OS$^{CDK2}$ cells cultures were supplemented with 500 µg/ml G-418 (G8168, Sigma). Human MCF10A were cultured in filtered DMEM-F12 (31331093, Thermo Fisher) supplemented with 10% FBS, 20 ng/ml EGF (AF-100-15, Peprotech), 0.5 mg/ml hydrocortisone (H4001, Sigma), 100 ng/ml cholera toxin (C8052, Sigma), 10 µg/ml insulin (I9278, Sigma), 100 U/ml penicillin, 100 µg/ml streptomycin, and 250 ng/ml fungizone at 37°C with 5% $CO_2$. Human RPE1 cells were cultured as previously described (*Grant et al., 2018*). Breast cancer cell lines were purchased from DSMZ (Germany) or ATCC (USA) with authentication certificates including STR profiling; all cell lines used in the study were free of mycoplasma contamination. Cell populations were enriched for $G_1$/S phase by incubating with 2.5 mM thymidine (T1895, Sigma) for 24 hr, for $G_2$ phase by incubating with 10 µM RO3306 (4181, Tocris Bioscience) for 16 hr, for M phase by incubating with 10 µM S-trityl l-cysteine (STLC) (2191/50, Tocris Bioscience) for 16 hr, and mitotic cells were then collected by shake-off. CHX (239763, Sigma), TMP (92131, Sigma), and DMSO (sc-358801, Insight Biotech) were used as indicated in the figure legends.

## Transfections

U2OS and RPE1 cells (5 × 10⁶) were electroporated (MPK5000, Neon Transfection System, Invitrogen) using 1150 V pulse voltage, 30 ms pulse width, and two pulses. U2OS$^{CDK2}$ and MCF10A cells (4 × 10⁴) were transfected using Lipofectamine 3000 Transfection Reagent (L3000001, Thermo Fisher) according to the manufacturer's instructions. MISSION microRNA Mimic *hsa-let-7a* (HMI0003, Sigma), miRNA Mimic Negative Control (ABM-MCH00000, abm), and Anti-miR miRNA Inhibitor (AM17000, Thermo Fisher) were co-transfected by Lipofectamine RNAiMAX Transfection Reagent (13778100, Thermo Fisher) according to the manufacturer's instructions. All analyses were carried out 24 hr post transfection.

## Live-cell fluorescence microscopy

Live-cell microscopy was performed using Olympus IX81 motorized inverted microscope, Orca CCD camera (Hamamatsu Photonics, Japan), motorized stage (Prior Scientific, Cambridge, UK), and 37°C incubation chamber (Solent Scientific, Segensworth, UK) fitted with appropriate filter sets and a 40× NA 1.42 oil objective. Images were collected in the 490 nm (Venus, sfGFP), 550 nm (mCherry), and 435 nm (CFP) channels using Micro-Manager software (*Edelstein et al., 2014*). Image analysis was performed using a customized plug-in tool in ImageJ (*Schindelin et al., 2012*), which calculates mean fluorescence intensity (MFI) by measuring average, background-subtracted gray values over regions of interest (ROIs) of defined diameter around manually selected points in the cell.

## Western blot

Western blot was performed as previously described (*Abdelbaki et al., 2020*). PageRuler Pre-stained Protein Ladder (26616, Thermo Fisher) was used. Primary antibodies were mouse anti-AURKA (1:1000; Clone 4/IAK1, BD Transduction Laboratories), rabbit anti-GFP (1:5000; ab290, Abcam), rabbit anti-GAPDH (1:4000; 2118S, CST), rabbit anti-mCherry (1:1000; ab167453, Abcam), mouse anti-Flag M2 (1:1000; F1804, Sigma). Secondary antibodies were rabbit (P044801-2) or mouse (P044701-2) HRP-conjugated (Dako, Agilent), used at 1:10000 dilution, and detection was performed via Immobilon Western Chemiluminescent HRP Substrate (WBKLS0100, Millipore) on an Odyssey Fc Dual-Mode Imaging System (LI-COR Biosciences).

## RNA extraction and RT-qPCR

RNA extracts were collected using Total RNA miniprep kit (T2010S, NEB). DNA was in-column digested with DNase I. Aliquots were stored with 5 µM EDTA at –20°C for a week or flash-frozen in dry ice and transferred at –80°C. A micro-volume spectrophotometer (NanoDrop Lite, VWR) was used to assess $A_{260}/A_{280}$ ratios of ~2.0 and $A_{260}/A_{230}$ ratios of 2.0–2.2. RT-qPCR was performed using Luna Universal One-Step RT-qPCR Kit (E3005S, NEB). 20 µl reactions using 200 nM primers and <100 ng RNA were run on ABI StepOnePlus Real Time PCR system following the manufacturer's instructions. Primers were designed at Eurofins Genomics. $\Delta C_t$ or $\Delta\Delta C_t$ method was used for relative quantifications accounting for the primer pairs amplification efficiency. Three technical replicates were performed in each biological replicate. Results shown as mean and SEM of three biological replicates. Assessment of DNA contamination, sequences of primers, validation of amplification efficiency of primer pairs, and RT-qPCR reaction conditions are provided (Appendix 1-figures 1-11, Appendix 1-table 1, Appendix 1-table 2). Checklist of MIQE guidelines (*Bustin et al., 2009*) can be found in *Supplementary file 1*.

## mRNA decay measurement

Cells were treated with 10 µg/ml actinomycin D (ActD) (10043673, Fisher Scientific) and RNA was isolated from cells at the indicated time points after inhibition of transcription. Target mRNA was quantified by RT-qPCR using the $\Delta\Delta C_t$ method with indicated reference targets and corresponding RNA extracts at 0 hr of ActD treatment as reference sample. Mean and SEM of three biological replicates are shown at each time point.

## Nascent chain immunoprecipitation

Transfected cells were treated with 0.1 mg/ml CHX for 15 min, then washed, centrifuged, and resuspended in ice-cold lysis buffer (100 mM Tris-HCl pH 7.5 [BP1757-100, Fisher Scientific], 500 mM LiCl [L7026, Sigma], 10 mM EDTA [10458654, Invitrogen], 0.1 mg/ml CHX, 0.1% Triton X-100 [28817.295, VWR], 100 U/ml RNasIn [3335399001, Merck], and cOmplete EDTA-free protease inhibitor cocktail [11836170001, Roche]) and incubated 15 min on ice. Lysates were cleared by centrifugation and supernatant was incubated with anti-Flag M2 magnetic beads (M8823, Sigma) overnight at 4°C rotating. The bound fraction was washed twice (10 mM Tris-HCl pH 7.5, 600 mM LiCl, 1 mM EDTA, 100 U/ml RNasin, 0.1 mg/ml CHX). Followed elution with 10 mM Tris-HCl pH 7.5, 600 mM LiCl, 1 mM EDTA, 100 U/ml RNasIn, 0.1 mg/ml puromycin (J67236.XF, Alfa Aesar), or with 3XFLAG peptide buffer (F4799, Sigma), for 30 min rotating at 4°C. RNA was purified (Monarch RNA Cleanup Kit, T2040L, NEB) from fractions and samples were stored as above. Aliquots of each fraction were mixed 1:1 with NuPAGE LDS Sample Buffer 4X (NP0007, Invitrogen) and 10 mM DTT (10197777001, Sigma), boiled 3 min at 95°C and stored at –20°C.

## 3'RACE

cDNA synthesis was performed using the Transcriptor Reverse Transcriptase (3531317001, Roche) with an oligo-dT anchor primer (5'-GACCACGCGTATCGATGTCGACTTTTTTTTTTTTTTTTV-3'). In the first round of PCR, an AURKA-specific forward primer (5'-TCCATCTTCCAGGAGGACCACTCTCTG-3') was used with a reverse primer for the oligo-dT anchor sequence (Anchor_R: 5'-GACCACGCGTATCGATGTCGAC-3'). In the second round of PCR (nested), a new AURKA-specific forward primer (5'-CGGGATCCATATCACGGGTTGAATTCACATTC-3') was used with Anchor_R. Nested PCR was performed using a 1:10 dilution of first PCR product as template. PCR product was visualized in agarose gels and imaged as above.

## Generation of ΔdPAS cell lines

Two guide RNAs (gRNA1: 5′-GGCCAACAATGAACAGATGG-3′ and gRNA2: 5′-GAGCAGGGGCTG AGAGGAGC-3′) were cloned into AIO-GFP (*Chiang et al., 2016*), a gift from Steve Jackson (Addgene plasmid #74119). The donor DNA template for homology-directed repair (HDR) was cloned into a separate vector expressing mRuby (mRuby-HDR). AIO-GFP-gRNAs and mRuby-HDR were co-transfected into U2OS cells and 48 hr after transfection GFP⁺mRuby⁺ cells were sorted at single-cell density into multiple 96-well plates for clonal expansion. Cell populations were individually screened for dPAS mutation by touch-down PCR. Mutants were confirmed by Sanger sequencing of the genomic locus.

## Cell counting kit-8 (CCK-8) assay

Cells were seeded into 96-well plates at a density of 4000 cells/well. CCK-8 (96992, Merck) was used according to the manufacturer's instructions and measurements were performed at the indicated time points. The $OD_{450}$ value was determined using a CLARIOstar Plus microplate reader (BMG LABTECH). Mean and SEM of three biological replicates shown for each time point.

## Colony formation assay in soft agar

A 0.6% base agarose was prepared in 6-well plates, and cells were seeded at density of 15,000 cells per well prior mixing with agarose to a final agarose concentration of 0.3%. Fresh medium was added every 3 d. Colonies were imaged and counted after 10 d using phase contrast microscopy under 4× magnification.

## Migration assay

A suspension of 50,000 cells was added to each well of a Culture-Insert 2 Well (IB-80209, Thistle Scientific Ltd) and grown into a monolayer. After insert removal, cells were washed and serum-free medium was added. At indicated time points, images of three different fields were acquired for every condition using phase contrast microscopy under 10× magnification. The percentage of cell-free area was calculated using Wound_healing_size_tool plugin (*Suarez-Arnedo et al., 2020*) in ImageJ and is shown as mean of three fields and SEM for three biological replicates.

## Statistical analyses

GraphPad Prism 9 (version 9.5.0; GraphPad Software Inc) and Microsoft Excel (version 16.72; Microsoft Corporation) were used to analyze data, generate graphs, and perform statistical analyses. Statistical parameters, including the sample size, the statistical test used, statistical significance (p-value), and the number of biological replicates, are reported in the figure legends or in the 'Materials and methods.'.

## Materials availability

Materials from this study are available from the corresponding author upon reasonable request.

# Acknowledgements

We thank past and present members of Lindon Lab for enriching discussions throughout the study. We are grateful to Chiara Marcozzi for advice on CRISPR/Cas9 and to Tim Weil, Adrien Rousseau, and Francesco Nicassio for insightful comments. Cartoon figures were created using https://www.biorender.com/. This work was supported by Biotechnology and Biological Sciences Research Council (BBSRC) (grant no. BB/R004137/1) to CL. RC is supported by David James Studentship from the Department of Pharmacology.

# Additional information

## Funding

| Funder | Grant reference number | Author |
|---|---|---|
| David James Trust | | Roberta Cacioppo |
| Biotechnology and Biological Sciences Research Council | BB/R004137/1 | Roberta Cacioppo |
| Scientific and Technological Research Council of Turkey | 112S478 | Hesna Begum Akman Taner Tuncer Ayse Elif Erson-Bensan |

The funders had no role in study design, data collection and interpretation, or the decision to submit the work for publication.

## Author contributions

Roberta Cacioppo, Conceptualization, Data curation, Formal analysis, Investigation, Methodology, Writing – original draft, Writing – review and editing; Hesna Begum Akman, Conceptualization, Software, Investigation, Methodology, Writing – review and editing; Taner Tuncer, Software; Ayse Elif Erson-Bensan, Conceptualization, Software, Investigation, Methodology, Project administration; Catherine Lindon, Conceptualization, Funding acquisition, Writing – original draft, Project administration, Writing – review and editing

## Author ORCIDs

Roberta Cacioppo ⓘ http://orcid.org/0000-0003-3048-9444
Ayse Elif Erson-Bensan ⓘ http://orcid.org/0000-0001-7398-9313
Catherine Lindon ⓘ http://orcid.org/0000-0003-3554-2574

Reviewer #1 (Public Review): https://doi.org/10.7554/eLife.87253.2.sa1
Reviewer #2 (Public Review): https://doi.org/10.7554/eLife.87253.2.sa2
Reviewer #3 (Public Review): https://doi.org/10.7554/eLife.87253.2.sa3
Author Response: https://doi.org/10.7554/eLife.87253.2.sa4

# Additional files

## Supplementary files

• MDAR checklist

• Supplementary file 1. MIQE guidelines checklist.

## Data availability

The data underlying this article are available in the article and in its supporting files.

The following previously published datasets were used:

| Author(s) | Year | Dataset title | Dataset URL | Database and Identifier |
|---|---|---|---|---|
| Karn T, Rody A, Schmidt M, Müller V, Holtrich U, Pusztai L, Kaufmann M | 2011 | A Clinically Relevant Gene Signature in Triple-Negative and Basal-Like Breast Cancer | https://www.ncbi.nlm.nih.gov/geo/query/acc.cgi?acc=gse31519 | NCBI Gene Expression Omnibus, GSE31519 |
| Graham KA, Rosenberg CL, Sebastiani P | 2010 | Histologically normal epithelium from breast cancer patients and cancer-free prophylactic mastectomy patients | https://www.ncbi.nlm.nih.gov/geo/query/acc.cgi?acc=GSE20437 | NCBI Gene Expression Omnibus, GSE20437 |

*Continued*

| Author(s) | Year | Dataset title | Dataset URL | Database and Identifier |
|-----------|------|---------------|-------------|------------------------|
| Tripathi A, King C, de la Morenas A, Perry KV, Burke B, Antoine G, Hirsch E, Kavanah M, Mendez J, Stone M, Gerry N, Lenburg M, Rosenberg C | 2007 | Gene expression abnormalities in histologically normal breast epithelium of breast cancer patients | https://www.ncbi.nlm.nih.gov/geo/query/acc.cgi?acc=GSE9574 | NCBI Gene Expression Omnibus, GSE9574 |
| Richardson A | 2006 | Human breast tumor expression | https://www.ncbi.nlm.nih.gov/geo/query/acc.cgi?acc=GSE3744 | NCBI Gene Expression Omnibus, GSE3744 |
| Liu R, Wang X, Chen G, Dalerba P, Gurney A, Hoey T, Sherlock G, Lewicki J, Shedden K, Clarke M | 2007 | The prognostic role of a gene signature from tumorigenic breast-cancer cells | https://www.ncbi.nlm.nih.gov/geo/query/acc.cgi?acc=GSE6883 | NCBI Gene Expression Omnibus, GSE6883 |
| Planche A, Bacac M, Provero P, Fusco C, Delorenzi M, Stehle J, Stamenkovic I | 2011 | Stromal molecular signatures of breast and prostate cancer | https://www.ncbi.nlm.nih.gov/geo/query/acc.cgi?acc=GSE26910 | NCBI Gene Expression Omnibus, GSE26910 |
| Schaefer C, Kemmner W | 2011 | Expression profiling of human DCIS and invasive ductal breast carcinoma | https://www.ncbi.nlm.nih.gov/geo/query/acc.cgi?acc=GSE21422 | NCBI Gene Expression Omnibus, GSE21422 |

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

## Appendix 1

**Appendix 1—table 1.** Sequences of primers and targets of primer pairs used in RT-qPCR.

| Primer pair | Sequence | Target |
| --- | --- | --- |
| mCherry_F<br>mCherry_R | CCGACATCCCCGACTACTTGAAGC<br>CACCTTGTAGATGAACTCGCCGTCC | mCherry ORF. |
| 3'UTR_F 3'UTR_R | GCCCGACAACCACTACCTGAGCTAC<br>GCTCAAGGATTTCTCCCCCTGCAC | Reporter Venus mRNAs containing AURKA 3'UTR. |
| Venus_F Venus_R | CTGACCCTGAAGCTGATCT GCAT<br>GGCGGACTTGAAGAAG | Venus ORF. |
| sfGFP_F sfGFP_R | GGCCCTGTCCTTTTACCAGACAACC<br>CATCCATGCCATGTGTAATCCCAGC | DHFR-sfGFP ORF. |
| 18S_F 18S_R | CTCAACACGGGAAACCTCAC CGCT<br>CCACCAACTAAGAACG | 18S ribosomal RNA* (NR_146146.1). |
| AURKA_Long_F<br>AURKA_Long_R | GGCGAAGCCTGGTAAAGCTG GCCT<br>CTTCTGTATCCCAAGCAAATCC | All AURKA 5'UTR splice variants[†] and only long 3'UTR APA isoform. Primer pair anneals to AURKA exon XI. |
| AURKA_Total_F<br>AURKA_Total_R | TGTAACAGAGGGAGCCAGGGACC<br>TGATGAATTTGCTGTGATCCAGGGGTG | All AURKA 5'UTR splice variants[†] and both 3'UTR APA isoforms. Primer pair anneals to AURKA exon XI. |
| AURKA_SLR_F<br>AURKA_SLR_Short_R | TGCTAGGCATGGTGTCTTCA AACA<br>GCTTTACCAGGCTTCG | All AURKA 5'UTR splice variants[†] and both 3'UTR APA isoforms. Primer pair anneals to AURKA exon XI. |
| AURKA_SLR_F<br>AURKA_SLR_Long_R | TGCTAGGCATGGTGTCTTCA AGAA<br>ACCCAATCAGGCCTAC | All AURKA 5'UTR splice variants[†] and only long 3'UTR APA isoform. Primer pair anneals to AURKA exon XI. |
| SDHA_F SDHA_R | TGGGAACAAGAGGGCATCTG CCAC<br>CACTGCATCAAATTCA | All SDHA transcript variants (NM_004168.4, NM_001294332.2, NM_001330758.2). Primer pair anneals to SDHA cDNA exons II–III. |

*Primer sequences from **Lin et al., 2016**.

[†]For accession numbers see **Cacioppo and Lindon, 2022**.

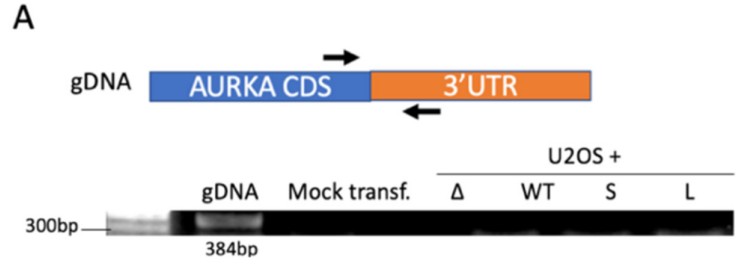

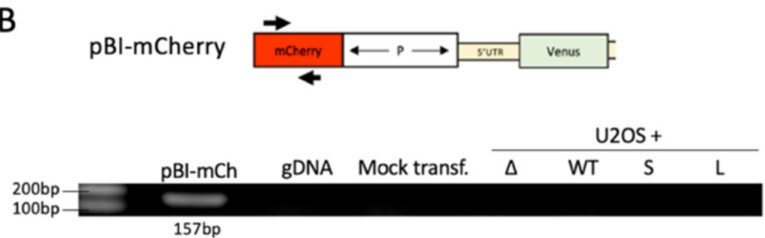

**Appendix 1—figure 1.** Assessment of DNA contamination of RNA extracts. (**A**) Lack of genomic DNA (gDNA) contamination in RNA samples was assessed by PCR using Aurora Kinase A (AURKA)-specific primers. RNA extracts from U2OS cells electroporated with the indicated reporters (**Figure 2A**) were used as template for the
*Appendix 1—figure 1 continued on next page*

*Appendix 1—figure 1 continued*

amplification reaction. An RNA extract from mock transfected U2OS cells used as negative control. gDNA from untransfected U2OS cells used as positive control. (**B**) Lack of plasmid DNA contamination in RNA samples was assessed by PCR using mCherry specific primers. RNA extracts from U2OS cells electroporated with the indicated reporters (*Figure 2A*) were used as template for the amplification reaction. An RNA extract from mock transfected U2OS cells and gDNA from untransfected U2OS cells used as negative controls. pBI-mCherry plasmid (pBI-mCh) used as positive control template.

**Appendix 1—figure 2.** Predicted targets on human genome of primer pairs used in this study. Analysis performed using UCSC In-Silico PCR tool (https://genome.ucsc.edu/cgi-bin/hgPcr). Genomic PCR product of SDHA primer pairs is above >850 bp length set for the in silico search.

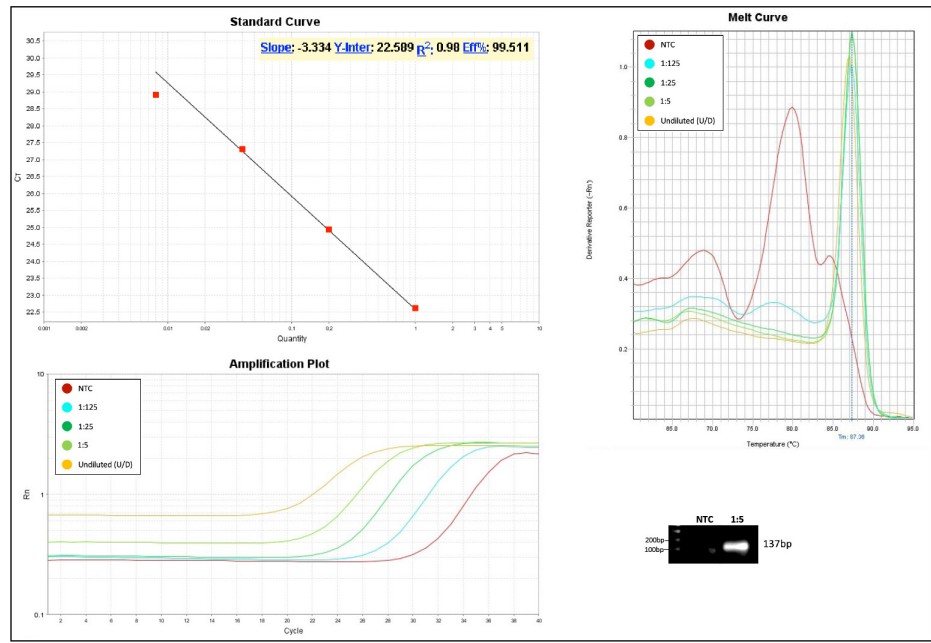

**Appendix 1—figure 3.** Validation of primer pairs used in RT-qPCR. Standard curve, melt curve, and amplification plot of amplification reactions of serial dilutions (1:5) are shown. Sample reactions were loaded on gel to validate amplicon size. NTC, non-template control.

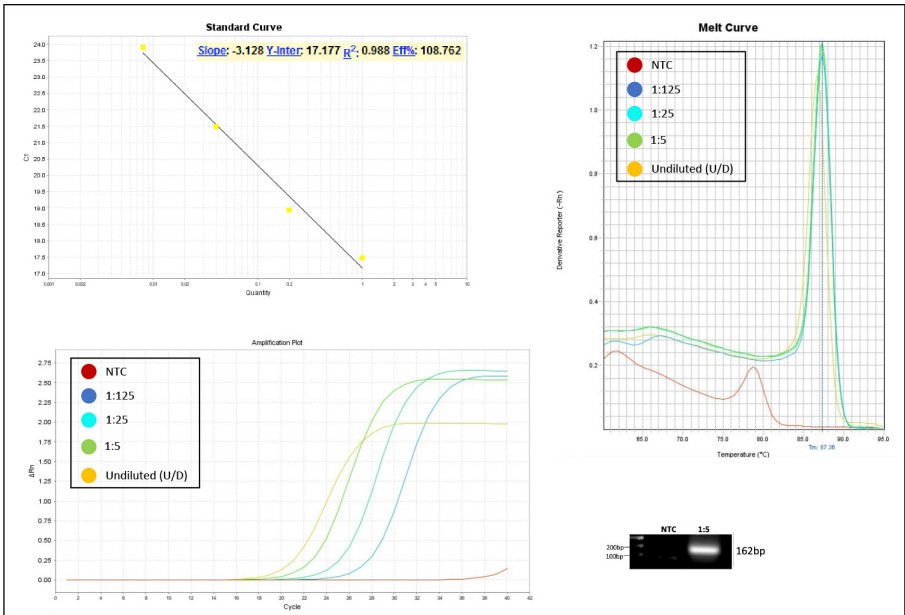

**Appendix 1—figure 4.** Validation of primer pairs used in RT-qPCR. Standard curve, melt curve, and amplification plot of amplification reactions of serial dilutions (1:5) are shown. Sample reactions were loaded on gel to validate amplicon size. NTC, non-template control.

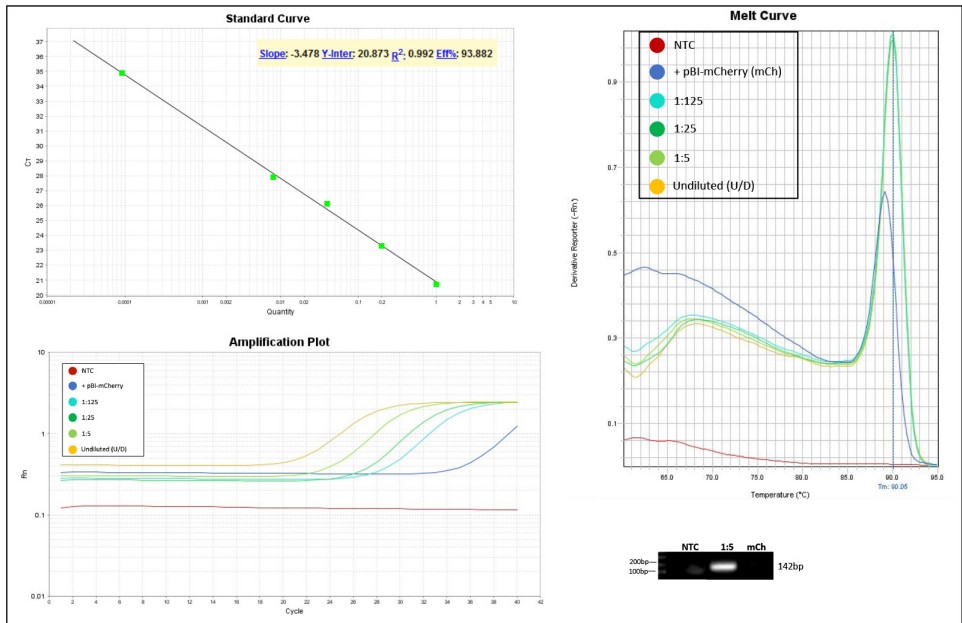

**Appendix 1—figure 5.** Validation of primer pairs used in RT-qPCR. Standard curve, melt curve, and amplification plot of amplification reactions of serial dilutions (1:5) are shown. Sample reactions were loaded on gel to validate amplicon size. The target was also amplified from RNA extracts of cells transfected with the pBI-mCherry construct as neg. control (mCh). NTC, non-template control.

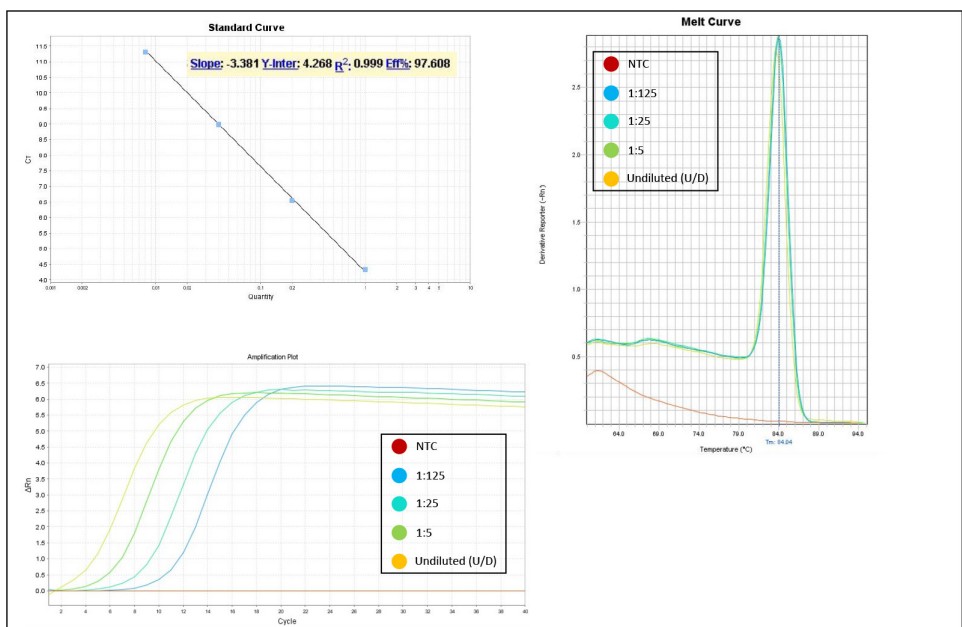

**Appendix 1—figure 6.** Validation of primer pairs used in RT-qPCR. Standard curve, melt curve, and amplification plot of amplification reactions of serial dilutions (1:5) are shown. NTC, non-template control.

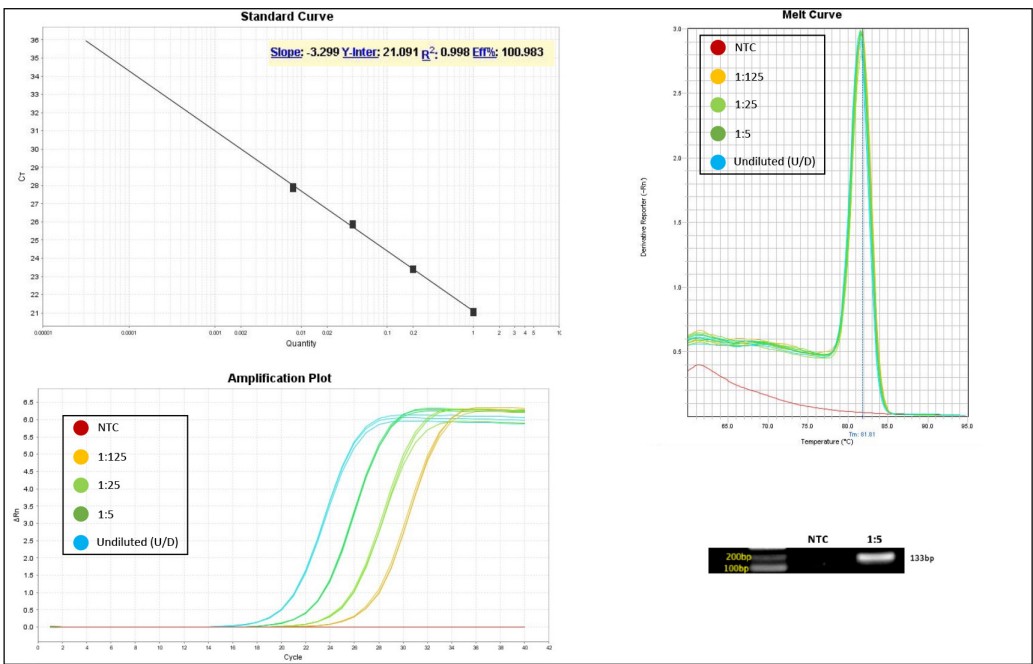

**Appendix 1—figure 7.** Validation of primer pairs used in RT-qPCR. Standard curve, melt curve, and amplification plot of amplification reactions of serial dilutions (1:5) are shown. Sample reactions were loaded on gel to validate amplicon size. NTC, non-template control.

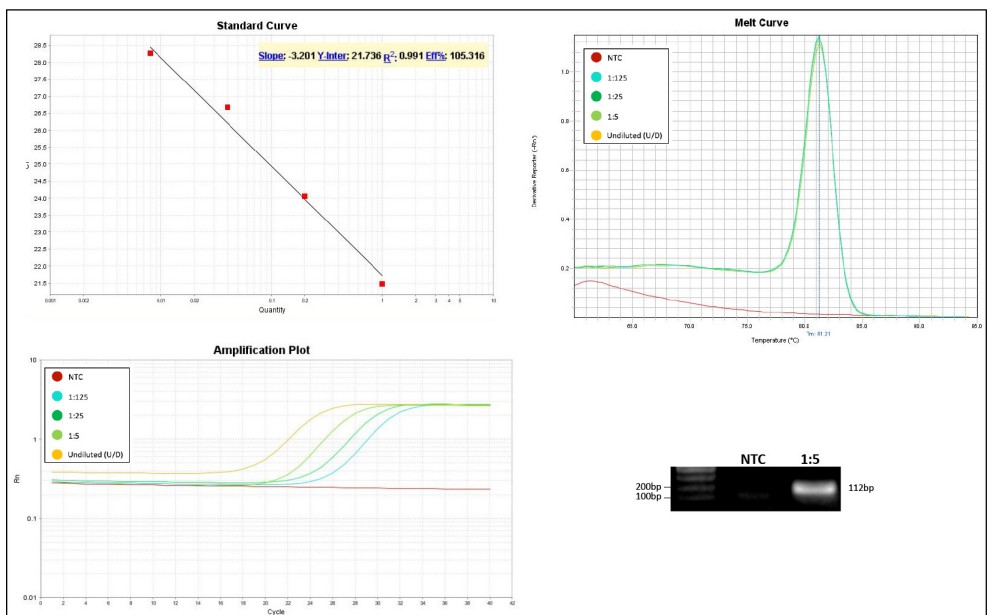

**Appendix 1—figure 8.** Validation of primer pairs used in RT-qPCR. Standard curve, melt curve, and amplification plot of amplification reactions of serial dilutions (1:5) are shown. Sample reactions were loaded on gel to validate amplicon size. NTC, non-template control.

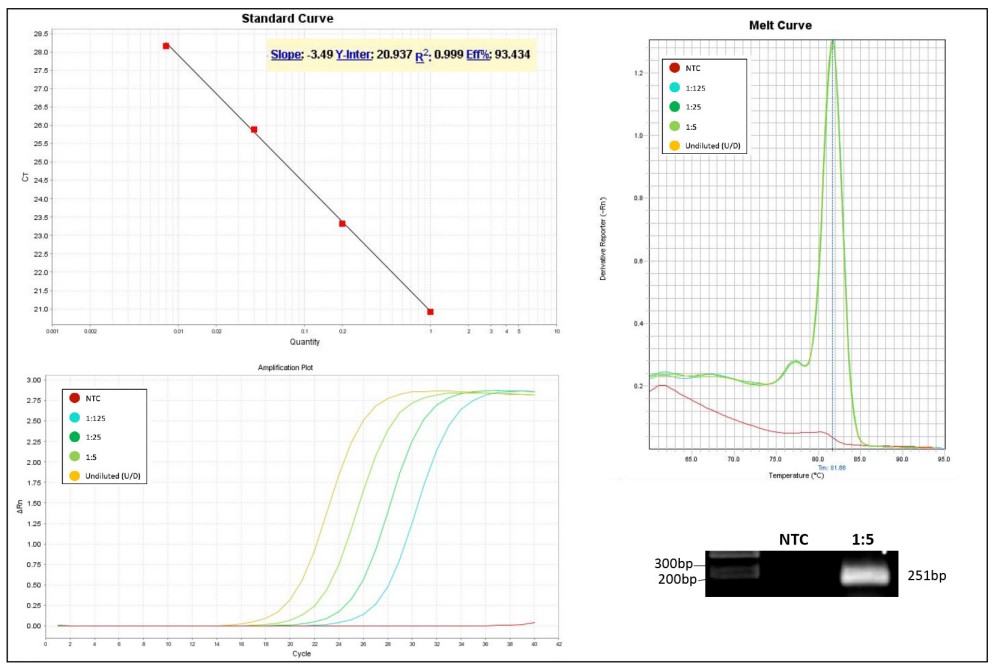

**Appendix 1—figure 9.** Validation of primer pairs used in RT-qPCR. Standard curve, melt curve, and amplification plot of amplification reactions of serial dilutions (1:5) are shown. Sample reactions were loaded on gel to validate amplicon size. NTC, non-template control.

## AURKA_SLR_F / AURKA_SLR_Short_R

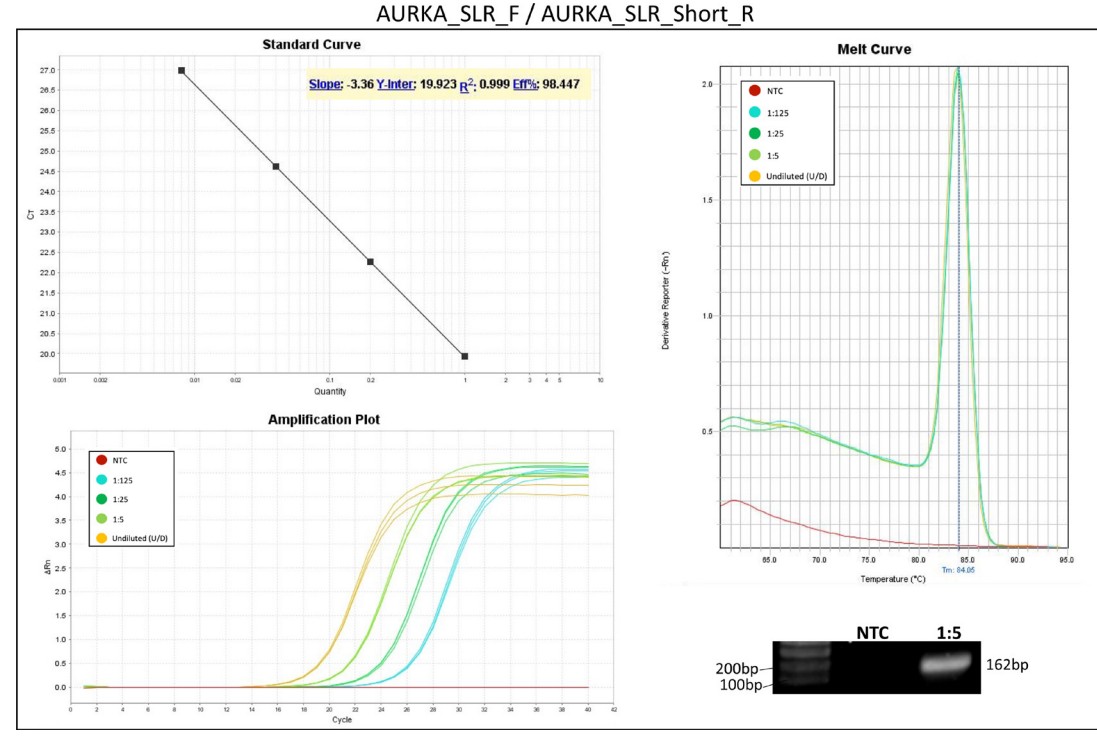

**Appendix 1—figure 10.** Validation of primer pairs used in RT-qPCR. Standard curve, melt curve, and amplification plot of amplification reactions of serial dilutions (1:5) are shown. Sample reactions were loaded on gel to validate amplicon size. NTC, non-template control.

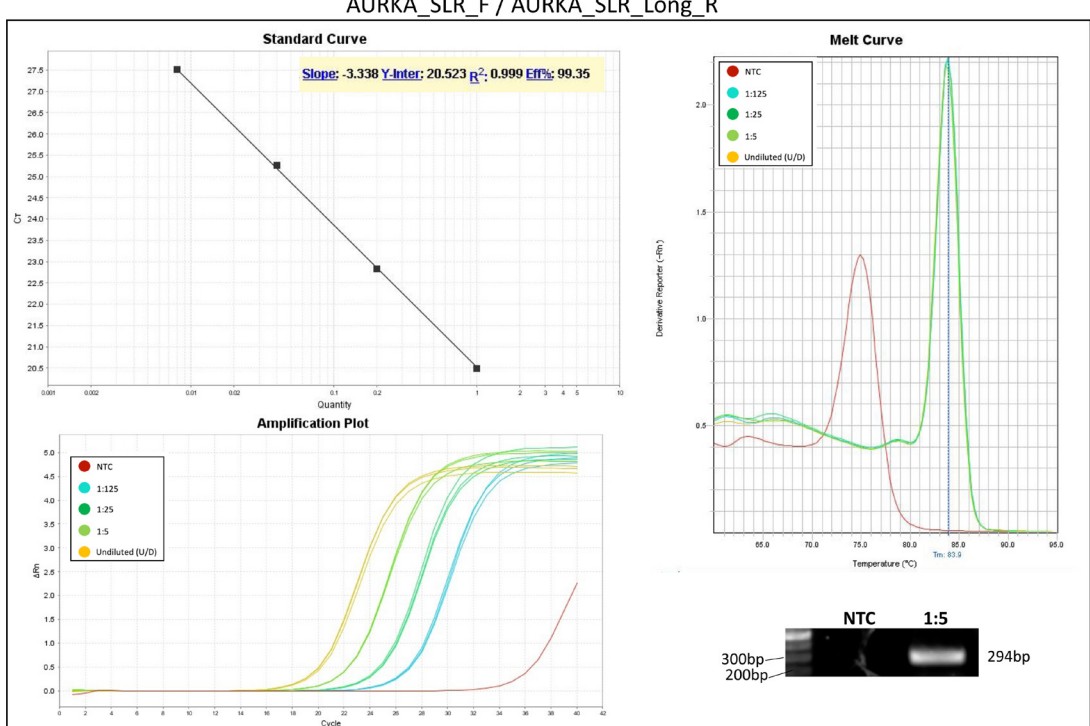

**Appendix 1—figure 11.** Validation of primer pairs used in RT-qPCR. Standard curve, melt curve, and amplification plot of amplification reactions of serial dilutions (1:5) are shown. Sample reactions were loaded on gel to validate amplicon size. NTC, non-template control.

**Appendix 1—table 2.** Reaction conditions and thermocycling parameters for one-step RT-qPCR.

| Reagent | Volume |
|---|---|
| Luna Universal One-Step Reaction Mix (2×) | 10 µl |
| Luna WarmStart RT Enzyme Mix (20×) | 1 µl |
| Forward primer (10 µM) | 0.4 µl |
| Reverse primer (10 µM) | 0.4 µl |
| Template RNA | <100 ng |
| Nuclease-free water | Up to 20 µl |
| MicroAmp Fast Optical 96-Well Reaction Plate with Barcode, 0.1 ml (Thermo Fisher, 4346906) | |

| Cycling step | Temperature (°C) | Time | Cycles |
|---|---|---|---|
| Reverse transcription | 55 | 10 min | 1 |
| Initial denaturation | 95 | 1 min | 1 |
| Denaturation extension | 95<br>–60 | 10 s<br>1 min | 40 |
| Melt curve | 60–95 | - | 1 |

