## [Editor Report · eLife assessment]

In this **important** study, the authors provide **compelling** evidence that the interplay between alternative polyadenylation (APA) of mRNA encoding Aurora Kinase A (AURKA) and *hsa-let-7a* miRNA governs AURKA protein levels. The authors show that short 3'UTR isoform of mRNA encoding AURKA is efficiently translated throughout the cell cycle, while the long 3'UTR isoform is suppressed by *hsa-let-7a* miRNA in a cell cycle-dependent manner. These findings delineate post-transcriptional mechanisms regulating AURKA expression that may be implicated in increase in AURKA protein that is frequently observed across a variety of cancers.

---

## [Referee Report · Reviewer #1 (Public Review)]

In this article, Cacioppo et al., report on a previously unappreciated mechanism of the regulation of Aurora Kinase A (AURKA) protein levels that is orchestrated via coordinated action of alternative polyadenylation of AURKA mRNA and hsa-let-7a miRNA. Moreover, it is proposed that this mechanism may play a major role in neoplasia. In support of their model, the authors demonstrate that short-to-long 3'UTR AURKA mRNA isoform ratio is elevated in triple negative breast cancer patients where it correlates with poor prognosis. The authors further generated reporters suitable for single cell live imaging that express different 3'UTR variants, which revealed highly variable ratios of short and long 3'UTR AURKA isoforms across different cell lines. This was followed by actinomycin D chase and nascent chain immunoprecipitation assays in U2OS osteosarcoma cells to demonstrate that while short and long 3'UTR AURKA isoforms have comparable stability, short 3'UTR AURKA isoforms appear to exhibit higher ribosome association which is indicative of higher translation activity. Furthermore, using an additional reporter assay which takes advantage of trimethoprim-based stabilization of highly unstable *E. coli* dihydrofolate reductase mutants Cacioppo et al., provide evidence that in contrast to the short 3'UTR AURKA mRNA isoform which appears to be constitutively translated throughout the cell cycle, long 3'UTR AURKA mRNA isoform is preferentially translated in the G2 phase. Further evidence is provided that suppression of long 3'UTR AURKA mRNA isoform is at least in part mediated by hsa-let-7a miRNA. Finally, the authors provide evidence that disrupting the expression of long 3'UTR AURKA mRNA isoform using CRISPR-based strategy, leads to overexpression of AURKA driven by the short 3'UTR isoform which is paralleled by an increase in cancer-related phenotypes.

Strengths: Overall it was thought that this study is of potentially broad interest inasmuch as it delineates a hitherto unappreciated mechanisms of regulation of AURKA protein levels, whereby AURKA is emerging as one of the major factors in neoplasia, including resistance to anti-cancer treatments. In general, it was thought that the author's conclusions were sufficiently supported by provided data. It was also thought that this study incorporates innovative methodology including single-cell expression sensors coupled with live cell microscopy and an assay to study translation in different phases of cell cycle without need for cell synchronization.

Weaknesses: Several relatively minor issues were observed regarding methodology and data interpretation. Namely, some inconsistencies between the models and/or cell lines that were used throughout the manuscript were noted. For instance, key experiments were performed almost exclusively in U2OS osteosarcoma cells, whereby triple negative breast cancer patient data were used to set the scientific foundation of the study. Considering potential differences in alternative polyadenylation between cell and tissue types, it was thought that investigation across the broader compendium of cell lines may be required for generalization of findings observed in U2OS cells. It was also found that the precise mechanisms underpinning the role of hsa-let-7a miRNA in regulation of AURKA protein levels remain largely obscure.

---

## [Referee Report · Reviewer #2 (Public Review)]

Cacioppo et al describe a mechanism of translation regulation of Aurora A, which is dependent on alternative polyadenylation. They suggest that altered expression of the resulting isoforms in cancers is at least partly responsible for elevated Aurora A levels, which in turn is known to indicate poor prognosis.

The authors exploit publicly available databases and patient data to highlight the correlation of increased abundance of the SHORT isoform (relative to the LONG one) and poor patient survival in TNBC, as well as breast and lung cancer.

In their thorough mechanistic study they use a number of reporters to assess the impact of alternative polyadenylation on mRNA stability and translation efficiency and explore whether this process accounts for cell-cycle-regulated expression of Aurora A. These reporters are carefully controlled and well explained. I particularly commend the authors for the clear graphical presentations of the reporters (eg fig 2A, fig 3D, fig 4A). Rigorous control experiments are performed to make sure that the reporters work and "report" what they are meant to do, and to show that previous findings can be reproduced in experiments based on the reporters (eg higher protein expression from the short 3' UTR APA isoform of CDC6 mRNA, targeting of MZF1 3'UTR by hsa-let-7a).

They show that translation of the longer isoform is subject to suppression by hsa-let-7a, while the shorter isoform is not. They attribute cell-cycle regulated expression of Aurora A at least in part to the suppression of translation of the LONG isoform in G1 and S.

In Figure 6 they address whether the APA-based regulatory mechanism alters Aurora A levels sufficiently to confer features associated with oncogenic transformation and overexpression of Aurora A. These data nicely tie together the observations in databases and the mechanistic part of the study.

The logic is clear and the conclusions are well supported by the data.

The authors state themselves that the impact of translation regulation on Aurora A levels in the cell cycle is an important but unanswered question. The evidence that suppression of translation of the LONG transcript contributes to the cell-cycle regulation of Aurora A is convincing, but the extent could be explored further. I wonder whether published genome-wide studies (eg PMCID 4548207, PMC3959127) have relevant data on the translation rate of Aurora A in the cell cycle.

In the paper this question is addressed in cells enriched in G1/S (Fig 6) and using the reporters (Fig 5). Having generated the ΔdPAS mutants, Aurora A levels could be easily assessed in each cell-cycle phase. The best way to do this would be sorting followed by immunoblotting.

The fact that Aurora A levels are reduced by a 6h treatment with 0.1 mg/ml CHX (Fig 6D) is interpreted as "AURKA expression in G1/S was reduced in the mutated cell lines when treated with CHX, indicating that translation of the short isoform is active in this phase" It is rather expected that using a translation inhibitor will stop the accumulation of a protein and so this experiment does not add much. A better approach to address the effect of the mutations on translation would be to add a proteasome inhibitor and follow accumulation of Aurora A, preferably not only in G1/S but also in other cell-cycle phases. Accumulation of the protein in this experiment would better reflect translation rates.

---

## [Referee Report · Reviewer #3 (Public Review)]

Summary:

This manuscript sheds light on the cell cycle-dependent post-transcriptional regulation of the oncogenic kinase AURKA. AURKA mRNA is subjected to alternative polyadenylation (APA), resulting in a short and a long 3'UTR isoform. While the ratio long/short isoform is important for AURKA expression and might impact cancer development, it is not unclear how this is regulated throughout the cell cycle. Translation and decay rate of the long isoform only are targeted by let-7a miRNA and in a cell-cycle dependent manner. In contrast, the short isoform is translated highly and constantly throughout interphase. Finally, depletion of the long isoform led to an increase in proliferation and migration rates of cells. In Triple Negative Breast Cancer, where AURKA is typically overexpressed, the short isoform is predominant and its expression correlates with faster relapse times of patients, suggesting that this mechanism might play an important role in this cancer.

Originality and novelty:

The originality of this work is to show the cooperation between APA and miRNA-targeting in controlling gene expression dynamics of AURKA during cell cycle. To investigate this mechanism, the authors have developed an interesting transient single-cell and biochemical assay to rapidly study mRNA-specific gene expression in a way that measures post-transcriptional events. This manuscript puts an emphasis on the cell cycle dependent expression control of AURKA at the translation level. However, the magnitude of the changes in mRNA levels throughout the cell cycle is even greater than that of the changes in translation. Therefore, it remains unclear whether translation really is that important in controlling AURKA expression during the cell cycle. Moreover, (i) AURKA regulation by miRNA is already known (Fadaka et al., Oncotarget 2020, Zhang et al., Arch Med Sci 2020, Yuan et al., Technol Cancer Res Treat 2019, Ma et al., Oncotarget 2015), (ii) the concept of cooperation between APA and translation already is not new (Sandberg et al., Science 2008, Mayr and Bartel, Cell 2009, Masamha et al. Nature 2015), and (iii) previous transcriptome-wide studies already suggested a cell-cycle dependent control of AURKA at the translation level (Tanenbaun et al., eLife 2015, translation efficiency ratio G2/G1 = 1.59) as well as the mRNA level (Krenning et al., eLife 2022). The impact of this manuscript could be increased by investigating (i) the mechanism of cell cycle-dependent regulation by let7a expression (i.e is there changes in let7a expression or activity during the cell cycle in this model) and (ii) the origin of AURKA APA dysregulation in cancer (could it be modulated by CFIm25? (Masamha et al. Nature 2015, Tamaddon et al. Sci Rep 2020)).

---

## [Author Response]

We thank the editors and reviewers for their assessment of our manuscript, and their agreement that we present compelling evidence for post-transcriptional regulation of AURKA through the 3’UTR.

In response to Reviewer 1, we acknowledge that much of our study is performed exclusively in U2OS cells, and that study of alternative polyadenylation in additional cell lines would serve to further generalize our findings. However, as U2OS are a well-known model cell line for cell cycle studies we believe our demonstration of cell cycle regulation of AURKA through its 3’UTR offers a depth of understanding that is perhaps of greater interest than confirming the existence of alternative AURKA 3’UTRs in additional cell lines, using our methods. We note that the recent rapid growth in RNA seq data resources allows easy confirmation of the broad existence of alternative polyadenylation events on a genome-wide scale. For example, AURKA-specific data extracted from a recent benchmark study of Nanopore long read RNA sequencing (Chen et al., 2021) clearly shows the existence of two distinct AURKA 3’UTRs differentially expressed between a number of different cancer cell lines. In addition, a recent study investigating the landscape of APA at single-cell resolution detected AURKA APA isoforms in HeLa and MDA-MB-468 cell lines (Wang et al., 2022). Their study further identifies AURKA among genes showing negative correlation between generalized distal polyA site usage index (gDPAU) and expression levels, meaning preference to use the proximal polyA site when expression levels increase, and include AURKA in the gene cluster showing slight increase in usage of the distal polyA site from G1 to M phase (Wang et al., 2022). Both studies are in support of the evidence presented in our manuscript.

We agree with Reviewer 2 that better information on translation rates would improve our understanding of the impact of translation regulation on AURKA levels. Some insight on the translation rate of AURKA in the cell cycle can be derived from inspection of the ribosome profiling dataset published by Tanenbaum et al., 2015. From their analysis, translation efficiency of AURKA mRNA in G2 is 1.59 times that in G1 and in G1 it is 0.69 times that in M phase, whilst in G2 it is 1.10 times higher than in M. Such data reveal a reversible increase in translation of AURKA mRNA, alongside other mitotic regulators, in preparation for M phase (Tanenbaum et al., 2015). These results are in accordance with our findings that translation rates contribute modestly to cell cycle changes in AURKA levels in normal cells, and we concur with Reviewer 3’s comment that the contribution of increased translation rate to AURKA levels at mitosis is less than the change in mRNA levels at this point in the cell cycle.

We think the significance of the regulatory mechanism we describe lies rather in the large effect it has on AURKA levels in interphase (when AURKA expression is normally repressed at both mRNA and translation rate). We hypothesise that it is interphase regulation that may be relevant to roles of AURKA in cancer (and to the association of APA with cancer) (Bertolin and Tramier, 2020; Naso et al., 2021). It is indeed the case that (i) AURKA regulation by miRNA, (ii) cooperation between APA and translation and (iii) cell-cycle dependent control of AURKA at the translation level, are already known. We believe the novelty of our study lies in drawing together these elements to provide new insight into AURKA regulation, using tools that allow similar investigation of other APA events, and contributing new ideas for future therapeutic interventions for disease proteins regulated via APA.

Bertolin and Tramier (2020), Cell. Mol. Life Sci https://link.springer.com/article/10.1007/s00018-019-03310-2Chen et al. (2021), bioRxiv https://www.biorxiv.org/content/10.1101/2021.04.21.440736v1Naso et al. (2021), Oncogene https://www.nature.com/articles/s41388-021-01766-wTanenbaum et al. (2015), eLife https://elifesciences.org/articles/07957Wang et al. (2022), PNAS https://www.pnas.org/doi/10.1073/pnas.2113504119